# Rating Quality of Diverse Time Series Data by Meta-learning from LLM Judgment

**Shunyu Wu**[1], **Dan Li**[1,*] **Wenjie Feng**[2,3], **Haozheng Ye**[1], **Jian Lou**[1], **See-Kiong Ng**[4]

[1]Sun Yat-sen University     [2]State Key Laboratory of AI Safety, Beijing, 100086
[3]University of Science and Technology of China     [4]National University of Singapore
```
{wushy88,yehzh8}@mail2.sysu.edu.cn,
{lidan263,louj5}@mail.sysu.edu.cn,
fengwenjie@ustc.edu.cn,   seekiong@nus.edu.sg
```

## Abstract

High-quality time series (TS) data are essential for ensuring TS model performance, rendering research on rating TS data quality indispensable. Existing methods have shown promising rating accuracy within individual domains, primarily by extending data quality rating techniques such as influence functions and Shapley values to account for temporal characteristics. However, they neglect the fact that real-world TS data can span vastly different domains and exhibit distinct properties, hampering the accurate and efficient rating of diverse TS data.

In this paper, we propose TSRating, a novel and unified framework for rating the quality of time series data crawled from diverse domains. TSRating leverages LLMs' inherent ample knowledge, acquired during their extensive pretraining, to comprehend and discern quality differences in diverse TS data. We verify this by devising a series of prompts to elicit quality comparisons from LLMs for pairs of TS samples. We then fit a dedicated rating model, termed TSRater, to convert the LLMs' judgments into efficient quality predictions by inferring future TS samples through TSRater's inference. To ensure cross-domain adaptability, we develop a meta-learning scheme to train TSRater on quality comparisons collected from nine distinct domains. To improve training efficiency, we employ signSGD for inner-loop updates, thus circumventing the demanding computation of hypergradients. Extensive experimental results on eleven benchmark datasets across three time series tasks, each using both conventional TS models and TS foundation models, demonstrate that TSRating outperforms baselines in terms of estimation accuracy, efficiency, and domain adaptability.

## 1 Introduction

Time series data arise ubiquitously across a broad range of domains, including healthcare (Morid et al., 2023), finance (Chan, 2004), industrial manufacturing (Wang et al., 2025), weather forecasting (Tian et al., 2025) and urban computing (An et al., 2025), which could exhibit vast distinctions in data characteristics from one domain to another (Cai et al., 2025a;b; Gong et al., 2025). Recent advances in large language model (LLM) techniques have unleashed new potential for leveraging these time series data to achieve promising performance in myriad time series tasks such as forecasting (Pan et al., 2024; Liu et al., 2024c; Shen et al., 2025), imputation (Fang et al., 2023; Wang et al.; Galib et al., 2024), and classification (Feofanov et al., 2025; Chen et al., 2024; Wen et al., 2024; Dong et al., 2024). There are roughly two categories of approaches, one finetunes or prompts off-the-shelf LLMs with format-converted time series data (Liu et al., 2024b; Gruver et al., 2023; Guan et al., 2025), and the other trains time series foundation models from scratch on vast time series datasets by adapting from LLM-inspired model architectures and training strategies (Ansari et al., 2024; Goswami et al., 2024; Rasul et al., 2023; Liu et al., 2024d; Woo et al., 2024; Wu et al., 2025). In both approaches, the quality of time series data is critical to model performance, as low-quality data can negate the

---

[*]Dan Li is the Corresponding Author.

[†]The code repository for TSRating is available at `https://github.com/clsr1008/TSRating`.

benefits of the bespoke technical advancements, even potentially leading to detrimental performance. In practice, time series datasets are frequently marred by pervasive quality issues, including missing values, sensor-failure-induced corruption, and irregular sampling (Fang et al., 2023; Wang et al.; Liu et al., 2024a; Zhang et al., 2024a). Consequently, it has become a pressing need to accurately and efficiently assess the quality of time series data crawled from diverse domains.

Existing attempts in this direction point out the necessity of developing tailored quality estimation methods for time series data to account for their unique temporal characteristics. For instance, Time-Inf (Zhang et al., 2024b) adapts the influence function (Hampel, 1974) to time series data, drawing inspiration from works like (Kunsch, 1984) (Infinitesimal robustness for autoregressive processes) and (Martin & Yohai, 1986) (Influence functionals for time series). Similarly, TimeShap (Bento et al., 2021) extends the Shapley value (Shapley, 1953), a cooperative game-theoretic attribution method, to time series data for quality assessment. However, current methods have largely neglected the fact that real-world time series data originate from diverse domains, rendering their effectiveness limited to a single domain. Furthermore, both influence function and Shapley value-based techniques face challenges in striking a balance between estimation fidelity and computational efficiency. Specifically, influence functions require computationally intensive Hessian and gradient calculations, while Shapley values incur exponential computational costs.

Recently, data quality estimation through the LLMs' judgments has emerged as a promising approach in the text field, offering accurate quality assessments by guiding LLMs with carefully designed prompts (Wettig et al., 2024; Gunasekar et al., 2023). The underlying rationale is that LLMs are proficient in understanding quality criteria for natural language, such as writing style, required expertise, factual accuracy, and educational value. As a result, LLMs can be successfully leveraged to assess the quality of text data.

Propelled by this success in the text data, it is tempting to explore whether LLMs can be exploited to determine the quality of diverse time series data. This approach has the potential to leverage LLMs' inherent knowledge of time series data from diverse domains that are acquired during their large-scale pretraining, thereby circumventing the prohibitive computational costs of calculating the influence function or Shapley value in a domain-by-domain manner. However, it remains under-explored whether LLMs can fully comprehend the key characteristics that underpin the quality rating for diverse time series data, and how to effectively steer them to distinguish between high- and low-quality time series data based on desired quality criteria.

**Our Work.** In this paper, we introduce TSRating, a novel and unified framework for rating the quality of time series data across diverse domains. First, we confirm that LLMs can indeed comprehend time series quality and discern differences in data quality along four criteria widely recognized as fundamental for characterizing time series data: trend, frequency, amplitude, and pattern. Next, we train a quality-prediction model, TSRater, which can efficiently rate the quality of large volumes of future time series data, incurring only minimal amortized computational cost once trained. To enhance adaptability across diverse domains, TSRater is trained via a meta-learning scheme on nine distinct domains, leveraging a large-scale and publicly available dataset released very recently by Time-MoE (Shi et al., 2024). We further alleviate training-time computation by using signSGD for inner-loop updates, thereby avoiding the computational burden of hypergradient calculations. Experimental results across eleven benchmark datasets and three representative time series tasks show that TSRating consistently surpasses state-of-the-art data selection baselines, including methods based on Shapley values and influence functions. We also perform data pruning experiments demonstrating that TSRating effectively pinpoints critical high-quality samples, whose removal significantly harms model performance. Additionally, ablation studies confirm the essential contribution of LLM-based criteria. Notably, a case study on time series foundation models further demonstrates TSRating's practical value: fine-tuning on high-quality subsets selected by TSRating significantly enhances generalization performance. These findings corroborate TSRating as an effective solution for time series quality assessment.

## 2 RELATED WORKS

**General Data Quality Estimation.** A range of data attribution methods has been proposed to assess the contribution of individual training samples to model behavior. Influence functions (Koh & Liang, 2017; Cook et al., 1982), originating from robust statistics, estimate the marginal impact

of a sample on model predictions or parameters by computing the product of gradients and the inverse Hessian matrix. Extensions such as TraceInf (Pruthi et al., 2020), ScaleInf (Schioppa et al., 2022), and TARK (Park et al., 2023) have scaled this approach to large foundation models, including LLMs (Grosse et al., 2023). Shapley value-based approaches (Ghorbani & Zou, 2019; Jia et al., 2019b; Huang et al., 2025a) offer a robust framework for estimating each sample's contribution through cooperative game theory. To reduce computational overhead, efficient approximations like KNN Shapley (Jia et al., 2019a) and gradient-based Shapley (Simon & Vincent, 2020) have been developed, making them applicable to larger datasets. Despite their theoretical fairness and model-agnostic applicability, these general methods often overlook the unique temporal properties inherent in TS data, such as continuity, temporal dependencies, and non-stationarity (Lu et al., 2025). This motivates the need for time series–specific quality evaluation approaches.

**Time Series Data Quality Estimation.** Several studies have attempted to adapt influence functions and Shapley values to the TS data. For example, Zhang et al. (2024b) proposes a time-aware influence function that preserves temporal dependencies and estimates the importance of TS segments. Wang et al. (2024a) further extends this to multivariate TS by introducing channel-level gradient approximations to capture the contribution of individual dimensions. Nayebi et al. (2023) employs windowing techniques (e.g., sliding windows) to decompose long sequences and apply Shapley computations at the block level. Cheng et al. (2025) has embedded Shapley computations into TS Transformer training loops to produce simultaneous forecasts and explanations. However, most of these approaches rely on assumptions of distributional homogeneity and fixed model representations, overlooking the fact that TS data often come from heterogeneous domains (e.g., healthcare, finance, weather). Such domain diversity introduces significant distribution shifts that can affect the generalizability of quality estimation methods (Sun et al., 2024). To address this limitation, our work proposes a unified and domain-adaptive solution to TS quality rating via LLM-integrated meta-learning.

**LLM-based Data Quality Estimation.** With the emergence of LLMs, recent work has explored their capabilities in data quality evaluation for text data. Qurating (Wettig et al., 2024) and Ask-LLM (Sachdeva et al., 2024) design prompt-based frameworks to evaluate aspects like educational value, factuality, and stylistic quality, enabling LLMs to serve as zero-shot quality judges. Other studies (Maini et al., 2024; Iskander et al., 2024; Shankar et al., 2024; Li et al., 2024) show that LLMs can annotate high-quality documents based on high-level human-aligned criteria. These efforts have validated LLMs' potential for cross-domain and interpretable data quality estimation, but are still restricted to textual and code-based data. The time series domain remains largely unexplored, despite the recent preliminary attempts to apply LLMs to time series tasks (Liu et al., 2024b;c; Jiang et al., 2025; Wu et al., 2026) and analysis (Jin et al., 2024; Zhou & Yu, 2024; Pan et al., 2024; Zhao et al., 2025). Inspired by the demonstrated capabilities of LLMs in understanding time series data, our work pioneers the application of LLM for time series quality rating.

## 3 THE PROPOSED TSRATING

### 3.1 OVERVIEW OF TSRATING

We propose TSRating, a unified framework for evaluating the quality across various criteria for time series data sourced from diverse domains. As shown in Figure 1, TSRating first prepares time series blocks using blocking techniques such as sliding windows. Then, LLMs are leveraged to assess each data block by considering four key characteristics of time series: trend, frequency, amplitude, and pattern. They are adopted as judgment criteria in the tailored prompts, which compare the quality of data blocks in a pairwise manner (Ouyang et al., 2022; Dubois et al., 2024). The comparison results are transformed into scalar values by the Bradley-Terry model. Then, these scalar values are applied to train a rating model, named **TSRater**, which maps the raw data representations to quality scores. To enable generalization across diverse time series domains, TSRater is trained via a meta-learning strategy. With a well-trained TSRater model, we can systematically select high-quality time series data to better facilitate downstream tasks such as forecasting and classification.

### 3.2 FORMULATION

Given a time series dataset $\mathcal{D} = \{(\mathbf{S}_t)\}_{t=1}^N$ with $N$ samples, each sample is defined as $\mathbf{S_t} = \{(x_1, y_1), (x_2, y_2), \ldots, (x_T, y_T)\}$, where $T$ is the number of time steps of the $t^{th}$ sample, $x_l \in \mathbb{R}^D$

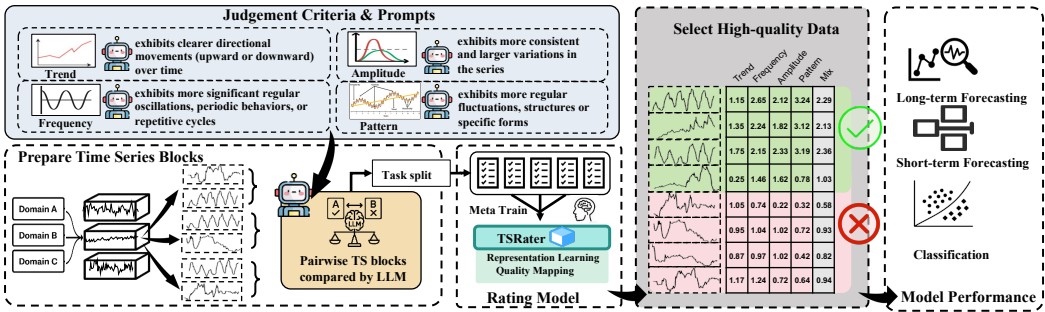

Figure 1: Overview of the proposed TSRating framework for diverse time series quality assessment.

is the measurement of the $l^{th}$ point, $y_l \in \mathbb{R}^C$ is the corresponding label, $D$ is the number of measurement channels, and $C$ represents the label dimensionality. Our goal is to evaluate the quality of each time series sample using the proposed rating framework-TSRating.

**Block-Level Scoring.** We first divide each time series sample $\mathbf{S_t} \in \mathbb{R}^{T \times D} \times \mathbb{R}^C$ into overlapping blocks. These data blocks are then assessed by LLM in a pairwise manner. For each block pair $\mathbf{B}_i$ and $\mathbf{B}_j$, LLM determines which block is better in terms of exhibiting more obvious quality criteria (trend, frequency, amplitude, and pattern), resulting in a binary preference annotation $m_{i \succ j} \in \{0, 1\}$:

$$m_{i \succ j} = \begin{cases} 1, & \text{if block } \mathbf{B}_i \text{ is preferred over } \mathbf{B}_j \\ 0, & \text{otherwise} \end{cases} \tag{1}$$

To enhance the reliability of these preferences, we compute the confidence score $p_{i \succ j}$ as the proportion of times $\mathbf{B}_i$ is preferred over $\mathbf{B}_j$ across multiple responses: $p_{i \succ j} = \frac{1}{M} \sum_{k=1}^{M} m_{i \succ j}^{(k)}$, where $M$ is the number of comparisons made. Then, these binary preference scores are transformed into scalar quality ratings using the Bradley-Terry model (Bradley & Terry, 1952): $p_{i \succ j} = \sigma(s(\mathbf{B}_i) - s(\mathbf{B}_j))$, where $\sigma(\cdot)$ is the sigmoid function and the scalar quality scores $s(\mathbf{B}_i)$ are estimated using maximum-likelihood estimation (MLE), which maximizes the probability of the observed pairwise judgments:

$$\mathcal{P} = \sum_{(\mathbf{B}_i, \mathbf{B}_j, p_{i \succ j}) \in \mathcal{J}} \Big[ p_{i \succ j} \log \sigma(s(\mathbf{B}_i) - s(\mathbf{B}_j)) + (1 - p_{i \succ j}) \log \sigma(s(\mathbf{B}_j) - s(\mathbf{B}_i)) \Big], \tag{2}$$

where $\mathcal{J}$ denotes the set of block comparisons. By maximizing $\mathcal{P}$, we obtain the most likely quality scores for each block.

In the case of multivariate time series, the LLM pairwise judgment is conducted channel by channel. The block preference score is the aggregation of channel scores: $s(\mathbf{B}_i) = \frac{1}{D} \sum_{d=1}^{D} s(\mathbf{B}_i^d)$, where $s(\mathbf{B}_i^d)$ is the preference score for channel $d$ in the $i^{th}$ block $\mathbf{B}_i$, and $D$ is the number of channels.

To provide unbiased estimate of the overall confidence $p_{i \succ j}$ between $\mathbf{B}_i$ and $\mathbf{B}_j$, we also swap the positions of $\mathbf{B}_i$ and $\mathbf{B}_j$ in the prompts and average the results from both orderings (Wang et al., 2023; Zeng et al., 2023b).

**Point-Level and Sample-Level Distribution.** With block-level quality scores, point-level quality scores are obtained by distributing the block-level quality scores back to individual time points: $s(x_i) = \frac{1}{|B(x)|} \sum_{\mathbf{B}_k \in B(x)} s(\mathbf{B}_k)$, where $B(x)$ is the set of blocks that contain the time point $x_i$. Next, to derive a sample-level quality score, the point-level quality scores for all time points in a sample are aggregated together: $s(\mathbf{S}) = \frac{1}{T} \sum_{i=1}^{T} s(x_i)$, where $\mathbf{S} = (\mathbf{x}, \mathbf{y})$ is a given time series sample with $\mathbf{x} = \{x_1, x_2, \ldots, x_T\}$.

### 3.3 JUDGMENT CRITERIA AND PROMPTS

Classic works in time series analysis (Cleveland et al., 1990; Hyndman & Athanasopoulos, 2018; Kazemi et al., 2019) have long treated trend, seasonality, periodicity, and other patterns as fundamental components for understanding and predicting time series behavior. Recent studies also incorporate such wave-based and signal-based elements to enhance the understanding of time series

data. For example, Cai et al. (2024) and Jing et al. (2026) propose benchmarks to evaluate LLMs' understanding of time series, using these properties as key dimensions. Similarly, Huang et al. (2025b) incorporates such features into the generation and forecasting of multi-domain time series. Besides, works like (Goswami et al., 2024) and (Woo et al., 2024) further encode these factors to improve the learning ability of machine learning models. Inspired by these findings, we adopt the following evaluation criteria to guide LLM's judgments, aiming to assess the quality of time series:

**Trend**: Evaluates the directional movement of data over time, identifying upward, downward, or stable trends. An obvious trend reduces the influence of random fluctuations and noise, suggesting the presence of a meaningful underlying factor driving the data.

**Frequency**: Examines the rate or periodicity of changes within the time series, capturing how often events or variations occur. Time series with clear frequency characteristics are often easier to interpret and forecast, as they reflect natural or structural processes (e.g., seasonal changes and cycles).

**Amplitude**: Measures the intensity or magnitude of fluctuations within time series, capturing the range of variation. A well-distributed amplitude in time series means that fluctuations or variations are significant and consistent, without being excessively dominated by noise or random disturbances.

**Pattern**: Assesses recurrent structures or sequences present in the data, such as seasonality, stationary variation, or their mixed cases. Recognizable patterns suggest that the data follows specific processes. Richer patterns in time series data enable more accurate forecasting, deeper insights, and more effective modeling by capturing underlying processes and reducing noise.

The prompt templates for the four criteria are provided in Appendix A, which are carefully crafted to guide the LLM in making consistent and interpretable comparisons for time series pairs. Through incorporating illustrative examples, explicit judgment criteria, and constraints to avoid spurious influences, we enhance the reliability of LLM-based annotations.

**Criteria Fusion.** The proposed LLM judgment criteria collectively generate criterion-wise quality scores for each time series block. While each criterion only reflects one particular aspect of the time series data, we combine them into a single final score that can reflect the overall quality of the time series blocks. The scores for each criterion are first normalized to a common scale and then aggregated as a final value for each block.

**Validation.** We conduct illustrative experiments to show whether LLMs can understand time series data with the designed prompts. On one hand, data blocks from a real-world dataset (Electricity as introduced in Section 4) are judged by GPT-4o-mini with the TSRating criteria, and we obtain the block-level quality scores. Then we visualize the highest and lowest-scored blocks selected in terms of each criterion in Figure 4 as shown in the Appendix B.1. The visualization results indicate that LLM's evaluation results align with human intuition.

On the other hand, we use a synthetic set where some blocks exhibit clearer and more structured temporal behaviors (trend, frequency, amplitude, and pattern), while others are intentionally corrupted with noise, making their temporal structures less distinct or difficult to discern. These synthetic blocks are then assessed by the LLM (GPT-4o-mini) in a pairwise manner to generate quality judgment results. The LLM achieves 94.5%, 92.25%, 98.75%, and 95.75% accuracy in identifying superior trend, frequency, amplitude, and pattern characteristics, respectively. The details of the synthetic data generation and experiments on other LLMs are presented in Appendix B.2. We additionally provide a comprehensive evaluation of the consistency and robustness of LLM-based judgments in Appendix G.

### 3.4 TSRater Training

**Model Architecture** As mentioned in Section 3.2, the TSRater model is designed to assign a scalar quality score to fixed-length time series blocks. Specifically, the input of the TSRater is a time series block, and the output is a scalar quality score according to the predefined quality criteria. Here, we train a separate TSRater for each quality criterion with two steps: mapping the time series from the original space to the representation space and mapping representations to scalar quality scores.

Representation Learning: A foundational time series model named MOMENT (Goswami et al., 2024) is adopted to extract essential temporal features from the input time series blocks. This model contains approximately 109 million parameters and is pretrained to capture critical time series patterns in the training data, making it a powerful encoder for time series representation. To maintain efficiency and stability during downstream training, the parameters of MOMENT are frozen.

Quality Mapping: Feature embeddings captured by the representation learning are then passed to a downstream Multi-Layer Perceptron (MLP), which maps the embedding vectors to scalar quality scores. The MLP consists of 3 fully connected layers with a hidden dimension of 256, striking a balance between expressiveness and computational efficiency. Each hidden layer is followed by LayerNorm and ReLU activation, and residual connections are added to improve training stability. The MLP is trained with the binary cross-entropy loss:

$$\mathcal{L}_\theta = \mathbb{E}_{(\mathbf{B}_i, \mathbf{B}_j, p_{i \succ j}) \in \mathcal{J}} \left[ -p_{i \succ j} \log \sigma(s_\theta(\mathbf{B}_i) - s_\theta(\mathbf{B}_j)) - (1 - p_{i \succ j}) \log \sigma(s_\theta(\mathbf{B}_j) - s_\theta(\mathbf{B}_i)) \right], \quad (3)$$

where $\mathcal{J}$ is the training dataset of LLMs' judgments, $s_\theta(\mathbf{B}_i)$ and $s_\theta(\mathbf{B}_j)$ are respectively the quality scores of blocks $\mathbf{B}_i$ and $\mathbf{B}_j$. The scores are parametrized by MLP model parameters $\theta$ to be trained. This training process aligns with the maximum likelihood estimation of Eq.(2).

**Meta-Learning**   To enhance the generalization across various domains and data sources, we adopt a model-agnostic meta-learning (MAML) strategy (Finn et al., 2017; Oreshkin et al., 2021) to train the TSRater. Rather than optimizing the model solely on a single dataset, meta-learning enables the TSRater to adapt quickly to new data distributions by learning from multiple related tasks. Each task corresponds to a specific dataset annotated with pairwise preferences for a particular quality criterion.

To construct the meta-training tasks of TSRater, we select 22 diverse data subsets from the Time-300B corpus (Shi et al., 2024). These subsets are drawn from 9 major domains, including energy, retail, finance, healthcare, transportation, weather, industry, synthetic, and other data, allowing TSRater to generalize across a wide variety of time series distributions and quality characteristics.

In each meta-training episode, we sample a set of inner tasks, where each task is split into a support set and a query set. The TSRater model is first adapted to each task using a few inner-loop updates by signSGD (Fan et al., 2021)). This choice inherently avoids computing hypergradients and their second-order derivatives. After adaptation, the model is evaluated on the corresponding query set, and the query loss is used to update the parameters of the meta-model. The objective function is:

$$\min_\theta \sum_{\mathcal{T}_i \sim \mathcal{T}} \mathcal{L}_{\mathcal{T}_i}^{\text{query}} \left( \theta - \alpha \cdot \text{sign} \left( \nabla_\theta \mathcal{L}_{\mathcal{T}_i}^{\text{support}}(\theta) \right) \right), \quad (4)$$

where $\theta$ is the shared model parameter, $\alpha$ is the inner-loop learning rate, and $\mathcal{L}_{\mathcal{T}_i}^{\text{support}}$, $\mathcal{L}_{\mathcal{T}_i}^{\text{query}}$ are the support and query losses for task $\mathcal{T}_i$, respectively. Appendix C provides a detailed description of the meta-learning algorithm used to train TSRater, as well as the model configurations, dataset details, and training settings. Besides, we also show results that demonstrate TSRater's ability to obtain robust quality estimation across various domains with minimal fine-tuning in Appendix C.4.

## 4 EXPERIMENTS

In this section, the effectiveness of the proposed TSRating framework is evaluated with three popular time series tasks, namely long-term forecasting, short-term forecasting, and classification. An extended case study on anomaly-aware data selection is provided in Appendix F to further demonstrate the extensibility of TSRating to task-specific criteria.

**Datasets and Baselines.** For Long-term forecasting, we utilize four widely adopted benchmark datasets: **Electricity** (Trindade, 2015), **Exchange Rate** (Lai et al., 2018), **Traffic** (tra), and **Weather** (wea). For Short-term forecasting, we adopt the **M4** dataset (Makridakis et al., 2018), focusing on its **Yearly**, **Monthly**, and **Daily** subsets. For Classification, we use four diverse datasets taken from the UCR/UEA archive (Ismail Fawaz et al., 2019; ucr): **MedicalImages**, **CBF**, **BME**, and **Handwriting**. For each dataset, we apply the meta-trained TSRater to assign quality scores. TSRater is first adapted via few-shot fine-tuning, and then used to score all data blocks. Detailed evaluation results on test sets are in Appendix C.4. We compare with four baselines: DataShapley (Ghorbani & Zou, 2019), KNNShapley (Jia et al., 2019a), DataOob (Kwon & Zou, 2023), TimeInf (Zhang et al., 2024b). Details of datasets and baselines are in Appendix D.1 and Appendix D.2.

**Time Series Tasks and Experimental Setup Details.** We consider three popular time series tasks: Short-Term and Long-Term Forecasting: We randomly sample 4,000 consecutive data points from each dataset and split them into training, validation, and test sets in a 7:1:2 ratio. A sliding window

Table 1: Comparison of data selection methods across forecasting and classification tasks on multiple datasets and models. Best results are **bolded**, second-best are underlined.

| Model | Method | Long-term (RMSE) | | | | Short-term (MAPE) | | | Classification (Accuracy) | | | |
|---|---|---|---|---|---|---|---|---|---|---|---|---|
| | | Elec. | ExRate | Traffic | Wea. | M4Y | M4M | M4D | MImg | CBF | BME | HW |
| Linear | Random | 1.601 | 0.356 | 0.979 | 0.665 | 1.705 | 1.208 | 1.672 | 0.390 | 0.294 | 0.427 | **0.053** |
| | DataOob | 1.539 | 0.318 | 0.761 | 0.638 | 1.949 | 1.133 | 1.779 | 0.457 | 0.318 | 0.260 | 0.028 |
| | DataShapley | 1.580 | 0.323 | 0.956 | 0.638 | **1.488** | 1.207 | **1.370** | 0.432 | 0.337 | 0.433 | 0.038 |
| | KNNShapley | **1.325** | 0.290 | 0.696 | 0.625 | 3.624 | 1.203 | 1.531 | 0.412 | 0.342 | **0.507** | 0.039 |
| | TimeInf | 1.391 | **0.272** | **0.609** | 0.616 | 1.966 | 1.178 | 1.463 | 0.428 | 0.320 | 0.500 | 0.042 |
| | TSRating | 1.390 | 0.275 | 0.683 | **0.611** | 1.577 | **1.112** | 1.409 | **0.459** | **0.361** | **0.507** | 0.049 |
| CNN | Random | 1.592 | 1.474 | 0.504 | 0.769 | 2.332 | 1.124 | 1.193 | 0.554 | 0.595 | 0.495 | 0.151 |
| | DataOob | 1.609 | 1.527 | 0.552 | 0.737 | 2.957 | 1.208 | 1.346 | 0.514 | 0.663 | 0.521 | 0.155 |
| | DataShapley | 1.529 | 1.598 | 0.475 | 0.767 | 2.553 | 1.117 | 1.159 | 0.550 | 0.593 | 0.564 | **0.159** |
| | KNNShapley | 1.620 | 1.500 | 0.553 | 0.763 | 2.099 | 1.312 | 1.674 | 0.533 | 0.621 | 0.539 | 0.131 |
| | TimeInf | 1.530 | 1.515 | 0.505 | 0.758 | 2.289 | 1.117 | **1.103** | **0.567** | 0.563 | 0.535 | 0.155 |
| | TSRating | **1.511** | **1.429** | **0.428** | **0.734** | **1.782** | **1.075** | 1.108 | 0.561 | **0.679** | **0.575** | **0.159** |
| PatchTST | Random | 0.406 | 0.222 | 0.361 | 0.474 | 4.146 | 2.044 | 1.985 | 0.561 | 0.470 | 0.475 | 0.124 |
| | DataOob | 0.416 | 0.226 | 0.362 | 0.558 | 3.982 | 2.060 | 1.958 | 0.571 | 0.489 | 0.489 | 0.141 |
| | DataShapley | **0.396** | **0.212** | 0.361 | **0.457** | 4.273 | 2.017 | 2.028 | 0.571 | 0.472 | 0.529 | 0.126 |
| | KNNShapley | 0.434 | 0.251 | 0.363 | 0.553 | 4.317 | 2.009 | 1.976 | 0.568 | 0.414 | 0.436 | 0.135 |
| | TimeInf | 0.415 | 0.220 | **0.351** | 0.510 | 4.029 | 2.019 | 2.023 | 0.522 | 0.467 | 0.504 | 0.131 |
| | TSRating | 0.397 | 0.213 | **0.351** | 0.467 | **3.901** | **1.957** | **1.863** | **0.572** | **0.511** | **0.536** | **0.156** |

approach then segments these time series into fixed-length blocks: 128 for long-term forecasting and 36 for short-term forecasting. Each block in the training set is treated as a standalone sample and assigned a quality rating via TSRating or other baseline methods. We select the top 50% of samples based on these ratings to train forecasting models, which are subsequently evaluated on the test set. Classification: We fix the block length at 100 and use TSRating's standard procedure to generate sample-level quality scores. In multivariate settings, each dimension is rated independently, and the final score is the average across all dimensions. We select the top 50% as training samples and evaluate classification performance on a held-out test set.

**Evaluation Metrics.** The quality of selected data is reflected in downstream model performance, namely, the more qualified the data is, the better the performance will be presented by the trained model. In this section, task-specific metrics are adopted. Root Mean Squared Error (RMSE) is taken for long-term forecasting, and Mean Absolute Percentage Error (MAPE) for short-term forecasting. A smaller RMSE or MAPE corresponds to a better prediction performance. Accuracy is used to measure classification performance. Higher classification accuracy indicates better model performance.

## 4.1 MAIN RESULTS

Table 1 presents the performance of different data selection methods across the three time series tasks with three representative model types: Linear, CNN-based, and Transformer-based (PatchTST) (Nie et al., 2022). In Appendix D.3, we also conduct extended experiments on additional models (e.g., TimeMixer (Wang et al., 2024b), DLinear (Zeng et al., 2023a), Non-stationary Transformer (Liu et al., 2022), etc.), showing TSRater's domain-agnostic effectiveness.

**Long-term Forecasting Task.** TSRating achieves the best RMSE in 6 out of 12 cases and ranks second-best in the remaining. Notably, it outperforms all baselines with CNN-based methods, reducing error by a significant margin over Random. Compared to TimeInf and DataShapley, TSRating achieves lower RMSE while maintaining higher consistency across datasets and architectures.

**Short-term Forecasting Task.** On the M4 datasets family, TSRating achieves the lowest MAPE in half of the evaluated cases. Compared to the baseline methods, TSRating shows superior adaptability across temporal granularities and robust performance across both simple and complex models. For example, on the M4-Yearly dataset with PatchTST, TSRating achieves the best performance with a MAPE of 3.901, outperforming the next best (DataOob) by a clear margin.

**Classification Task.** For classification, TSRating yields the best accuracy in 10 out of 12 cases. In the CNN and PatchTST groups, TSRating consistently boosts performance, particularly achieving clear improvements over DataShapley and TimeInf on CBF and BME. These results underscore TSRating's ability to guide model training towards more informative samples.

**Computational Efficiency.** To demonstrate TSRater's practical efficiency, we evaluate its runtime performance on the **Time-300B benchmark** (Shi et al., 2024), which spans 9 domains and 22 subsets. Each method is required to complete an evaluation across all subsets under the same hardware setup to ensure fairness. The total runtime (in seconds) is reported in Table 2.

TSRater's total runtime is comparable to DataOob and TimeInf and substantially faster than DataShapley. KNNShapley costs the least runtime due to its simplicity, at the sacrifice of evaluation accuracy. Importantly, TSRater's amortized per-dataset cost is significantly reduced because the meta-trained model can be reused: given a new dataset, only the lightweight few-shot tuning and inference steps (about 200 seconds) are required, whereas competing methods must be retrained or re-evaluated from scratch on every new dataset. This efficiency gain primarily stems from the strong generalization ability of the meta-trained rater, which avoids rerunning the full evaluation pipeline for each new dataset. As dataset size and diversity increase, this reusability will translate into even greater efficiency advantages over baseline methods. More analyses on computational efficiency are presented in Appendix D.4.

Table 2: Runtime comparison of TSRater and four baselines on the Time-300B benchmark. The total token usage is approximately 14.3 million tokens, averaging around 0.65 million tokens per dataset. * indicates estimates from the original paper.

| Method / Component | Time (s) |
|---|---|
| DataShapley | 210,000* |
| KNNShapley | 152 |
| DataOob | 4,785 |
| TimeInf | 4,938 |
| TSRater (Total) | 4,687 |
| – LLM Judgments | 4,167 |
| – Meta-training | 323 |
| – Few-shot Tuning | 55 |
| – Inference | 142 |

## 4.2 DATA PRUNING

The data pruning experiment aims to assess the ability of quality evaluation methods to identify the most influential data, which is critical to improving model performance by focusing on the most relevant information. To achieve this, we remove samples from the training set based on their quality scores in descending order and showcase the model's performance changes based on the same test set. As shown in Figure 2, with high-quality data removed, TSRating shows a faster performance decline compared to baselines. For example, for the PatchTST model trained by the Traffic dataset, its RMSE values increase by more than 0.03 after removing the top-40% high-quality samples selected according to TSRating's quality scores. Whereas, methods like KNNShapley and DataOob experience RMSE increases in the range of 0.01 to 0.02, respectively. Similarly, with the other two datasets (M4 for short-term forecasting and CBF for classification), the model trained with samples selected according to TSRating also exhibits the fastest overall performance decline compared to other methods when high-quality data is removed from the training set.

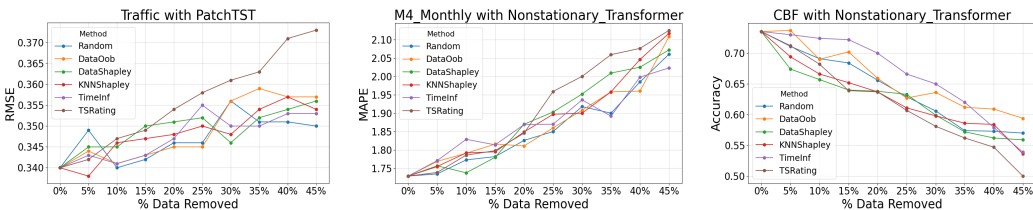

Figure 2: Data pruning comparison. Time blocks with the highest ratings are iteratively removed, and the performance degradation is measured. Higher RMSE and MAPE in Left and Middle, and lower accuracy in Right subfigures indicate more accurate rating methods.

## 4.3 ANALYSIS

**Generalization of Meta-rater.** A central motivation of TSRating is to develop a meta-trained rater that generalizes across datasets, obviating the need to retrain a separate rater for each one. To validate

this, we conduct an ablation study on the Electricity dataset, comparing three variants: (1) Meta-rater, the rater trained via meta-learning across multiple datasets. (2) Single-dataset rater (same dataset), a rater trained only on the Electricity dataset. (3) Single-dataset rater (other dataset), a rater trained on the Weather dataset and directly applied to Electricity. These variants are then used to evaluate and select high-quality samples from the Electricity dataset for downstream model training.

The RMSE results under five downstream models are reported in Table 3. It is observed that the Meta-rater achieves nearly identical accuracy to the rater trained directly on the Electricity dataset, demonstrating that meta-learning preserves task-specific information. In contrast, the rater trained on a different dataset (Weather) suffers from significant performance degradation when evaluated on Electricity, highlighting the importance of cross-dataset generalization.

Table 3: Effect of meta-learning on the Electricity dataset. The meta-rater matches the performance of a single-dataset rater trained on Electricity, while substantially outperforming a rater trained on another dataset (Weather). Best RMSE results are **bolded**.

| Method | Linear | CNN | PatchTST | iTransformer | TimeMixer |
|---|---|---|---|---|---|
| Meta-rater | **1.390** | 1.511 | **0.397** | **0.300** | 0.345 |
| Single-dataset rater (same dataset) | 1.471 | **1.497** | 0.398 | 0.306 | **0.332** |
| Single-dataset rater (other dataset) | 1.556 | 1.602 | 0.418 | 0.310 | 0.382 |

**Ablation on Quality Criteria.** For the ablation study, we compare the results of using individual criteria for scoring against the performance achieved by mixing all of them together. Table 4 demonstrates that while single criteria can perform well on certain datasets, the performance is unstable. For example, the "amplitude" criterion achieves the best result on the Weather dataset but performs the worst on the Traffic dataset. When all criteria are fused, the performance improves, and it consistently outperforms the individual criteria. This ablation study underscores the importance of incorporating multiple criteria to achieve a comprehensive evaluation. We further discuss potential adaptive aggregation strategies for combining multiple criteria in Appendix E. More ablation studies on framework components and experiment configurations can be found in Appendix D.5 and Appendix D.6.

Table 4: Ablation study results comparing the performance of individual criteria and the combined *fusion* approach across four long-term forecasting datasets using the model TimeMixer. The metric is RMSE. Best results are **bolded**, and second-best are underlined.

| Criteria | Weather | Electricity | Exchange | Traffic |
|---|---|---|---|---|
| trend | 0.475 | 0.366 | 0.239 | 0.291 |
| frequency | 0.476 | 0.392 | **0.216** | 0.294 |
| amplitude | **0.418** | 0.336 | 0.242 | 0.320 |
| pattern | 0.500 | 0.396 | 0.234 | **0.265** |
| fusion | 0.433 | **0.318** | 0.223 | 0.285 |

Table 5: Forecasting performance on the Weather dataset using TSRater with different TSFM encoders for data selection.

| Encoder | Linear | CNN | PatchTST | iTransformer | TimeMixer |
|---|---|---|---|---|---|
| MOMENT | 0.611 | 0.734 | 0.467 | 0.444 | 0.433 |
| Chronos | 0.613 | 0.735 | 0.464 | 0.447 | 0.428 |
| TimeGPT | 0.609 | 0.738 | 0.479 | 0.438 | 0.435 |

**Ablation on TSFM Encoders.** To investigate the dependence of TSRater on the choice of different types of time series foundation model (TSFM) encoders, we conduct ablation experiments comparing three representative TSFM encoders: MOMENT, Chronos, and TimeGPT. Using the Weather dataset, TSRater employs each encoder to perform data selection, followed by evaluation of downstream forecasting performance on five widely used models. All experimental settings are kept consistent with the main experiment. The results in Table 5 show comparable performance across different

TSFM encoders, indicating that TSRater's effectiveness is not strongly dependent on a particular encoder architecture. This suggests that TSRater's gains primarily stem from its ability to learn quality assessments guided by LLM supervision, rather than relying solely on pretrained TSFM representations.

**Alternative LLMs.** We also explored Claude-3.5 and Gemini-2.0 for LLM judgment to evaluate how alternative LLMs would affect the performance of TSRating. We conduct these experiments on the classification CBF dataset. As shown in Table 6, results based on three LLMs (GPT-4o-mini, Claude-3.5, and Gemini-2.0) are similar across different task-specific models. GPT-4o-mini is suggested as a cost-effective choice due to its lower API cost without sacrificing much performance. We further examine a multi-LLM ensemble strategy in Appendix D.8.

Table 6: Comparison of TSRating using different LLMs (GPT-4o-mini, Claude-3.5, Gemini-2.0) on the classification CBF dataset. The metric is classification accuracy (%). Best results are **bolded**.

| LLM | Linear | CNN | Informer | Nonstationary Transformer | PatchTST |
|---|---|---|---|---|---|
| GPT-4o-mini | 0.360 | **0.679** | 0.648 | 0.733 | **0.531** |
| Claude-3.5 | **0.367** | 0.660 | 0.657 | 0.704 | 0.528 |
| Gemini-2.0 | 0.362 | 0.651 | **0.697** | **0.736** | 0.522 |

### 4.4 CASE STUDY ON FINETUNING TIME SERIES FOUNDATION MODELS

To further illustrate the effectiveness of TSRating, we conduct a case study by finetuning three representative TSFMs, i.e., Time-MoE (Shi et al., 2024),Time-LLM (Jin et al., 2023) and MO-MENT (Goswami et al., 2024), using data of varying quality rated by TSRater as finetuning datasets. As shown in Figure 3, all three models achieve lower forecasting errors (MSE) when trained on higher-quality samples, even with reduced data volume. This case study highlights the practical value of TSRating in real-world scenarios, where training resources are limited and prioritizing high-quality data can significantly enhance model performance. Additional results using the MAE metric are provided in Appendix D.7, which further corroborate these findings.

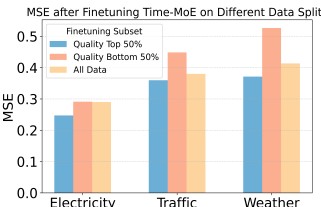 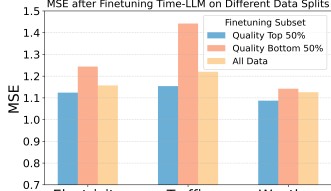 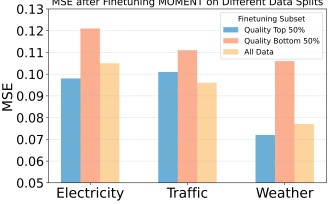

Figure 3: Mean Squared Error (MSE) on test sets after finetuning different time-series foundation models using varying portions of training data across three datasets (**Left:** Time-MoE, **Middle:** Time-LLM, **Right:** MOMENT). For each dataset, the model is fine-tuned using either the top 50% highest-quality data, the bottom 50%, or the full dataset. Models finetuned on higher-quality subsets consistently achieve lower MSEs, demonstrating the effectiveness of quality-based data selection.

## 5 CONCLUSION

In this paper, we present TSRating, a unified framework for rating the quality of diverse time series data. We first confirm that LLMs can effectively understand and compare the quality of time series through pairwise judgments. We train a generalizable rating model via meta-learning to adapt efficiently across multiple domains. Extensive experiments demonstrate that TSRating consistently outperforms existing baselines in quality estimation accuracy and efficiency. Future work may explore more adaptive ways to integrate quality criteria and segment time series data.

ACKNOWLEDGMENTS

We would like to thank anonymous reviewers and area chairs for their constructive comments and efforts in improving our paper. This work was supported in part by the Open Funding Programs of State Key Laboratory of AI Safety. We also gratefully acknowledge the computational resources and support provided by the National Supercomputer Center in Guangzhou.

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

APPENDIX

In this appendix, Section A provides full details of all prompt templates designed for our TSRating framework; Section B presents additional experiments to demonstrate LLMs' capability in evaluating the quality of time series data; Section C provides further details of the TSRater model, including the derivation of the meta-learning algorithm, the composition of datasets for meta-learning, detailed settings of the training process, and additional experiments for the meta-learned TSRater; Section D presents further experiment details and extended experiment results; Section E investigates adaptive aggregation of quality criteria; Section F provides an extended case study on anomaly-aware data selection; Section G presents further evaluation of LLM judgment consistency and robustness; Section H discusses potential limitations of the current work; Section I clarifies the scope of LLM usage in our work. Finally, our code is provided in the supplementary material and can also be accessed via `https://github.com/clsr1008/TSRating`.

## A    FULL DETAILS OF PROMPTS DESIGNED FOR OUR TSRATING

To elicit high-quality pairwise judgments from large language models (LLMs) for discerning differences in time series data quality, we carefully design prompt templates for each quality criterion: **trend**, **frequency**, **amplitude**, and **pattern**. In particular, the design rationale behind our prompt templates can be summarized as follows, aligning with common prompt design principles (Wettig et al., 2024):

**Explicit Instruction:** Each prompt begins with a clear and concise definition of the target criterion (e.g., trend or frequency), emphasizing what constitutes a "well-defined" signal and providing examples to guide LLM behavior.
**Positive and Negative Examples:** Inspired by in-context learning, each prompt includes both positive (well-defined) and negative (poorly-defined) examples to provide contrastive supervision. These examples serve as implicit demonstrations, helping the LLM better understand the boundaries of each quality criterion and reduce ambiguity in its judgments.
**Instructional Constraints:** Prompts specify aspects that should not influence decisions, such as time series length or order, to mitigate spurious correlations and enforce focus on the intended characteristic. However, these instructions alone do not fully eliminate biases. For instance, we observe that models like GPT-4o-mini still exhibit positional bias, favoring one option due to its location in the prompt. To counteract this, we swap the positions of two blocks and average the scores across both orderings.
**Forced Choice Format:** We use a pairwise comparison setup with two labeled options (Option A and B), and ask the LLM to select one. This relative judgment setup aligns with Bradley-Terry modeling and avoids challenges associated with absolute scoring.
**Minimal Output Constraint:** The final instruction emphasizes single-word responses (A or B), encouraging deterministic and parsable outputs.

These prompts are designed to elicit consistent and interpretable pair-wise comparisons from LLMs across a wide range of time series data. Full prompt templates for all four criteria are presented below.

---

**Prompt Template for Trend**

**Compare two time series and choose the one that exhibits a more significant and well-defined trend**, e.g., a clear directional movement (upward or downward) over time that is sustained across the series, with minimal noise, anomalies, or random fluctuations.

For example:
A time series with a steady upward trend, such as [10, 15, 20, 25, 30], would be considered significant and well-defined.
Conversely, a time series with frequent random spikes or drops, such as [10, 50, 20, 5, 30], is less likely to exhibit a well-defined trend.
A flat time series with little to no change, such as [10, 10, 10, 10, 10], would generally be considered to lack a significant trend.

---

Aspects that should NOT influence your judgment:
The source or origin of the time series data.
The length of the time series.
The order in which the time series are presented.

The time series may have similar characteristics, but you should still make a relative judgment and choose the label of the preferred time series.

[Option {label_a}]
... {series_a} ...
[Option {label_b}]
... {series_b} ...

Now you have to choose between either {label_a} or {label_b}. Respond only with a single word.

---

### Prompt Template for Frequency

**Compare two time series and choose the one that exhibits more significant and well-defined frequency or cyclical patterns**, e.g., regular oscillations, periodic behavior, or repetitive cycles that are consistent across the series, with minimal noise or randomness.

For example:
A time series with a consistent cyclical pattern, such as [10, 20, 10, 20, 10, 20], would be considered significant and well-defined.
Conversely, a time series with irregular peaks or inconsistent cycles, such as [10, 50, 20, 5, 30], is less likely to exhibit a well-defined cyclical pattern.
A flat time series with little to no change, such as [10, 10, 10, 10, 10], would generally be considered to lack significant frequency or cyclical behavior.

Aspects that should NOT influence your judgment :
The source or origin of the time series data.
The length of the time series.
The order in which the time series are presented.

The time series may have similar characteristics, but you should still make a relative judgment and choose the label of the preferred time series.

[Option {label_a}]
... {series_a} ...
[Option {label_b}]
... {series_b} ...

Now you have to choose between either {label_a} or {label_b}. Respond only with a single word.

---

### Prompt Template for Amplitude

**Compare two time series and choose the one that exhibits more significant and well-defined amplitude**, e.g., consistent and large variations in the range of values across the series, reflecting strong oscillations or signal intensity with minimal noise or randomness.

For example:
A time series with a large and consistent amplitude, such as [0, 10, -10, 10, -10, 10], would be considered significant and well-defined.
Conversely, a time series with small or inconsistent amplitude, such as [1, 2, 1, 2, 3, 2], is less likely to exhibit well-defined amplitude behavior.
A flat time series with minimal changes, such as [5, 5, 5, 5, 5], would generally be considered to lack significant amplitude.

Aspects that should NOT influence your judgment :
The source or origin of the time series data.
The length of the time series.

The order in which the time series are presented.

The time series may have similar characteristics, but you should still make a relative judgment and choose the label of the preferred time series.

[Option {label_a}]
... {series_a} ...
[Option {label_b}]
... {series_b} ...

Now you have to choose between either {label_a} or {label_b}. Respond only with a single word.

---

### Prompt Template for Pattern

**Compare two time series and choose the one that demonstrates a clearer and more consistent pattern**, exhibiting regular fluctuations, trends, or cycles, while avoiding excessive noise, random fluctuations, or sudden irregularities. Look for data that reflects some form of underlying structure, such as trend, seasonality, or cyclical behavior.

For example:
Trend Pattern: A time series with a clear and steady upward or downward trend, such as [5, 8, 11, 14, 17] and [43, 36, 29, 22, 15, 8, 1], demonstrates a well-defined, consistent direction.
Cyclic Pattern: A time series showing periodic cycles, like [30, 25, 20, 25, 30, 35, 40, 35, 30] and [1, 3, 1, 3, 1], repeating every few steps, suggests cyclical behavior over time.
Stationary Pattern: A time series where the values fluctuate around a stable mean without a clear upward or downward trend, such as [10, 12, 11, 10, 13] and [10, 10, 10, 10, 10], shows consistent, predictable variation.
Mixed Pattern: A time series that combines trends with cyclical or seasonal behavior, such as [10, 15, 20, 25, 30, 28, 25, 23, 28, 33, 38, 43, 41, 38, 36], where both a rising trend and periodic fluctuations are visible, would indicate a complex but structured pattern.

On the other hand, avoid time series with the following characteristics:
Random or Irregular Fluctuations: A time series with large, unpredictable jumps, such as [10, 50, 20, 5, 80], is erratic and lacks consistent patterns.
Noise-Dominant Data: A time series filled with random noise or frequent outliers, like [20, 5, 15, 100, 3], introduces significant unpredictability that makes it hard to discern any underlying trend or pattern.
Missing or Incomplete Patterns: A time series with large gaps or inconsistent segments, such as [15, ?, ?, 30, 40], is incomplete and might mislead pattern identification.

Remember, focus on time series that show clear, repeatable patterns of behavior. Even if the series contains some minor fluctuations, the overall trend, cycle, or stationary nature should be identifiable.

Aspects that should NOT influence your judgment:
The source or origin of the time series data.
The length of the time series.
The order in which the time series are presented.

[Option {label_a}]
... {series_a} ...
[Option {label_b}]
... {series_b} ...

Now you have to choose between either {label_a} or {label_b}. Respond only with a single word.

## B  ADDITIONAL EXPERIMENTS FOR DEMONSTRATING LLMS' CAPABILITY IN DISCERNING TIME SERIES DATA QUALITY

This section presents additional experiments to demonstrate LLMs' capability in discerning time series data quality based on our proposed TSRating criteria. The following contains two parts: (1)

Section B.1 provides experiments and analysis based on real-world time series data; (2) Section B.2 provides more assessments based on synthetic datasets with controlled characteristics.

## B.1 VISUALIZATION RESULTS AND ANALYSIS ON REAL-WORLD TIME SERIES DATA

To illustrate that LLMs can understand time series characteristics, we applied TSRating to evaluate all blocks in the real-world Electricity dataset based on the four criteria: trend, frequency, amplitude, and pattern. We then visualized the five highest and five lowest-scored blocks for each criterion to gain insight into how well the model aligns with human intuition. These blocks are selected based on their quality scores, which reflect how well they exhibit the respective criteria. By comparing these blocks visually, we could examine whether LLM can accurately capture meaningful trends, variations, and patterns in the time series data. The visualization results of the Electricity dataset are shown in Figure 4.

**Analysis of Trend criterion.** According to Subfigure 4 (a) and (b), the highest-scoring blocks exhibit clear and consistent directional movement, either showing an upward or downward trend within a single phase. These blocks are characterized by smooth and predictable changes, aligning with the definition of the trend as a consistent shift over time. On the other hand, the lowest-scoring blocks for trend are erratic and lack any distinct direction. These blocks display irregular fluctuations, with no discernible increase or decrease over time periods, confirming LLM's ability to distinguish between blocks with clear trends and those with noisy or unpredictable patterns.

**Analysis of Frequency criterion.** According to Subfigure 4 (c) and (d), blocks with high scores demonstrated repetitive oscillations or cycles, indicating a strong periodic behavior. These blocks consistently followed a rhythmic pattern, reinforcing the idea of frequency as the recurrence of regular intervals within the data. In contrast, the bottom-scoring blocks in terms of frequency displayed chaotic, non-repetitive behavior, with no clear cyclic pattern. These blocks lacked the regularity expected for a high-frequency score, further highlighting TSRating's capacity to detect structured versus irregular data patterns.

**Analysis of Amplitude criterion.** According to Subfigure 4 (e) and (f), the highest-scoring blocks exhibited large and significant fluctuations in magnitude, clearly showing variations that were both notable and consistent across time. These blocks typically had an amplitude around 1000, with considerable changes in intensity, fitting the definition of amplitude as the extent of variation. Conversely, the lowest-scoring blocks for amplitude were flat and showed minimal variation, with their amplitude ranging between 300 and 400, exhibiting only small, insignificant fluctuations. This behavior confirmed that TSRating effectively identifies blocks with substantial intensity changes and differentiates them from those with insignificant variation.

**Analysis of Pattern criterion.** According to Subfigure 4 (g) and (h), the blocks with high scores contained recognizable periodic, seasonal or stationary structures, where repetitive sequences were easily identified. These blocks were structured and predictable, aligning with our design and description of the pattern criterion of specific shapes or forms within the data. On the other hand, the bottom-scoring blocks lacked coherent structures, appearing as random noise or outliers. These blocks showed no repeating sequences or consistent organization, confirming TSRating's ability to accurately identify data blocks with recognizable patterns from those without clear structures.

**Analysis of Combined Criteria.** In addition, we visualize the scoring results obtained by combining all criteria. Specifically, we present the highest- and lowest-ranked blocks according to the aggregated quality score on the Electricity dataset in Figure 5. To further assess the generality of this behavior under more realistic and diverse temporal patterns, we conduct the same visualization on the Weather dataset, which contains more realistic noise and irregular fluctuations. The corresponding results are shown in Figure 6. These combined-criteria visualizations further demonstrate that TSRating can consistently identify globally informative and less informative time series segments across real-world domains.

These visualizations provide clear evidence that TSRating is capable of identifying and differentiating the key characteristics of time series data quality. The LLM's ability to effectively score and differentiate data blocks based on these criteria aligns well with human expectations, demonstrating the LLM's understanding of time series concepts.

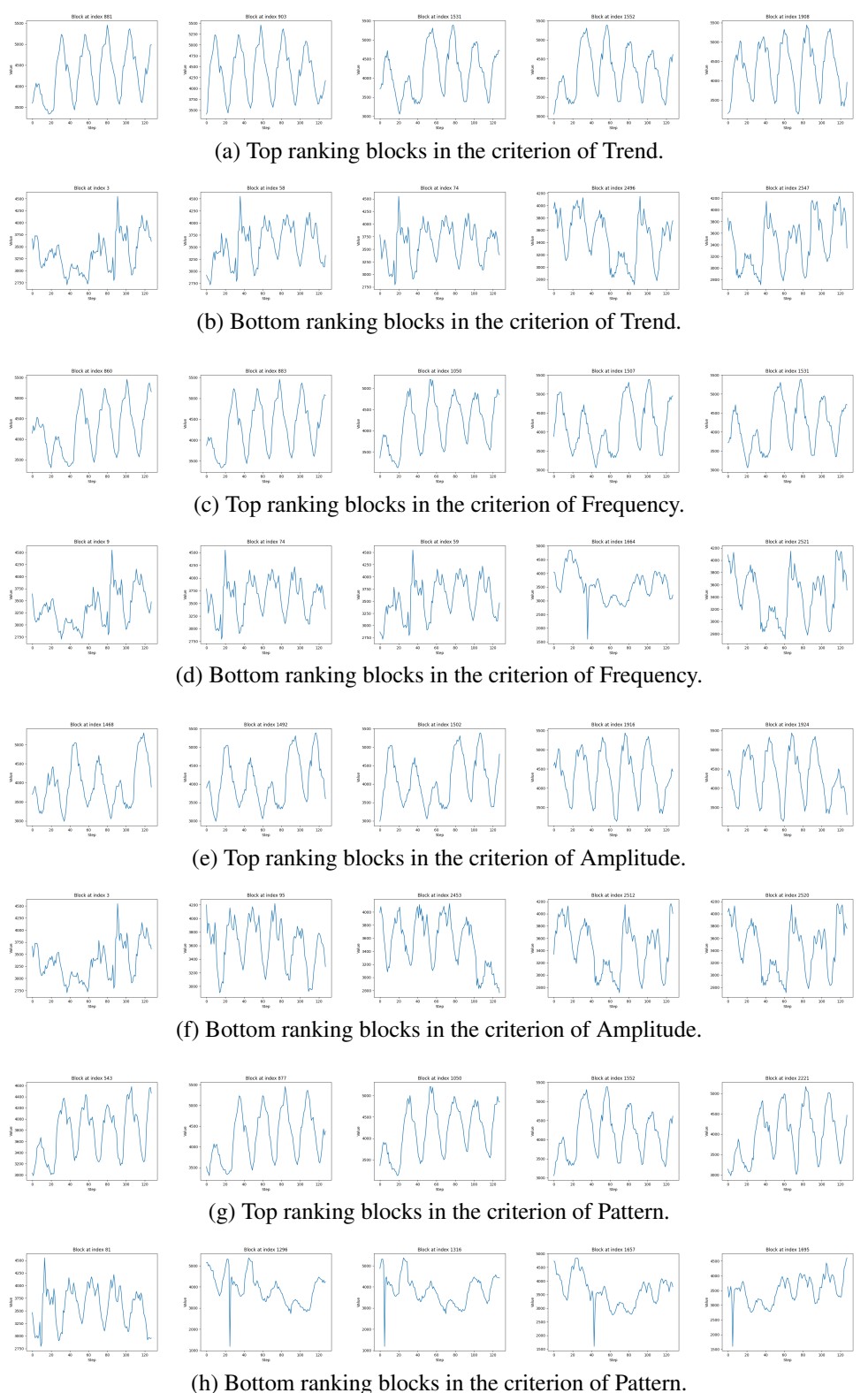

Figure 4: Visualization of the top and bottom ranking blocks in the Electricity dataset for each criterion (Trend, Frequency, Amplitude, and Pattern) with TSRating.

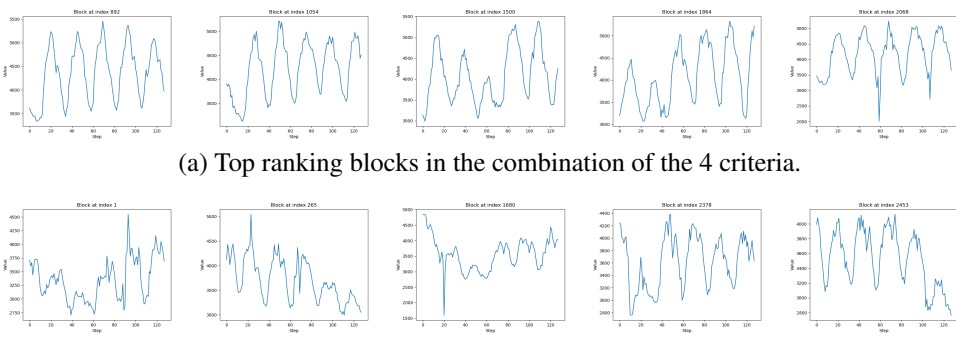

(a) Top ranking blocks in the combination of the 4 criteria.

(b) Bottom ranking blocks in the combination of the 4 criteria.

Figure 5: Visualization of the top and bottom ranking blocks in the Electricity dataset for the combination of four criteria with TSRating.

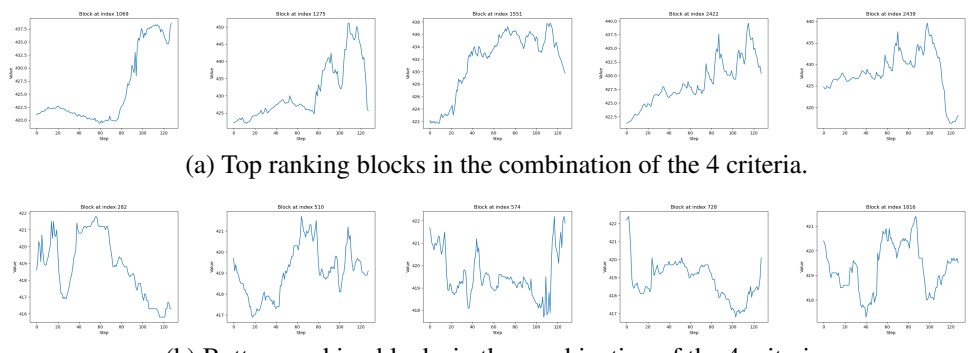

(a) Top ranking blocks in the combination of the 4 criteria.

(b) Bottom ranking blocks in the combination of the 4 criteria.

Figure 6: Visualization of representative high- and low-ranked time-series blocks on Weather dataset under the combined TSRating criteria. Compared with the Electricity dataset that mainly shows visually ideal samples, we further expand the visualization to include examples with more realistic noise and irregular fluctuations. As shown in (a), blocks ranked higher by TSRating generally exhibit clearer trends, seasonality, and structural regularity. In contrast, the lower-ranked blocks in (b) tend to contain weaker or unstable periodicity, irregular movements, or disrupted structures. These results support our claim that TSRating can effectively distinguish samples with more meaningful temporal patterns even under real-world complexity.

## B.2 EVALUATION RESULTS AND ANALYSIS ON SYNTHETIC TIME SERIES DATA

We further synthesize a validation dataset comprising 800 pairs of purposely constructed time series samples with designated characteristics. Each pair contains one high-quality and one low-quality block based on a specific criterion as defined in our framework.

**High-quality Data Synthesis.** High-quality Data are characterized by clear features according to a particular criterion. For the **Trend** criterion, we use low-noise linear functions and also create variations by superimposing multiple linear functions with different slopes, capturing a broader range of trends. For the **Frequency** criterion, we use low-noise sine functions with fixed frequencies and amplitudes to produce regular and repetitive oscillations, mimicking periodic signals. For the **Amplitude** criterion, we employ low-noise sine functions with varying amplitudes, representing smooth and consistent changes in signal strength. For the **Pattern** criterion, we combine multiple sine functions of different frequencies and phases along with linear functions to generate complex, periodic patterns with varying shapes.

**Low-quality Data Synthesis.** Conversely, low-quality data lack distinguishable features for the criterion and are generated by adding Gaussian noise to the time series for each of the four criteria. This noise is specifically designed to obscure the underlying patterns, making it difficult to discern

any clear trends, cycles, or periodicity. In addition, the noise is carefully tuned to ensure that it adequately masks the key characteristics of the high-quality blocks, thereby creating a challenge for the LLM in distinguishing between high and low-quality data. The validation dataset includes 200 high-quality samples and 200 low-quality samples per criterion, resulting in 200 pairs for each. The code for generating the validation dataset is available in our released code repository.

**Results and Analysis.** We evaluate the capability of three state-of-the-art LLMs in distinguishing time series quality across defined criteria based on our designed prompts. As shown in Table 7, all three LLMs—GPT-4o-mini, Claude-3.5, and Gemini-2.0—demonstrate strong performance across the four criteria, with accuracies consistently above 89%. This confirms that different LLMs can effectively understand and distinguish key characteristics of time series data based on our prompts, even without fine-tuning. Notably, **Claude-3.5** achieves near-perfect performance, reaching a perfect 100% on *Frequency* and 99.75% on *Amplitude*, highlighting its exceptional consistency and precision. **Gemini-2.0** excels on the *Pattern* criterion with a perfect 100% accuracy, indicating its strength in understanding complex temporal compositions, although its performance on *Frequency* is relatively lower (89.25%). **GPT-4o-mini** demonstrates solid and stable performance across all four criteria, with accuracies exceeding 92%, making it a reliable option. In addition to its balanced accuracy, the significantly lower API cost of GPT-4o-mini further motivates its adoption in our main experiments.

Overall, the consistently high accuracies across different LLMs reinforce the reliability of our prompt design, and demonstrate that LLMs can serve as effective tools for evaluating time series data quality in a criterion-specific manner.

Table 7: Performance comparison of various LLMs on the synthetic dataset, measured by identification accuracy for each quality criterion.

| LLM | Trend | Frequency | Amplitude | Pattern |
|---|---|---|---|---|
| GPT-4o-mini | 0.9450 | 0.9225 | 0.9875 | 0.9575 |
| Claude-3.5 | 0.9950 | 1.0000 | 0.9975 | 0.9675 |
| Gemini-2.0 | 0.9575 | 0.8925 | 0.9900 | 1.0000 |

## C    COMPREHENSIVE DETAILS OF TSRATER DERIVATION AND MODEL TRAINING

This section provides comprehensive details of the TSRater derivation and training. The following contains four parts: (1) Section C.1 provides a detailed derivation of the meta-learning algorithm used to train TSRater; (2) Section C.2 provides details of training dataset construction for meta-learned TSRater; (3) Section C.3 provides detailed settings and hyperparameter selection during TSRater training; (4) Section C.4 provides further experimental results for meta-learned TSRater.

### C.1    DETAILED DERIVATION OF TRAINING ALGORITHM FOR META-LEARNED TSRATER

Algorithm 1 outlines the meta-training procedure for the proposed TSRater, based on the meta-learning paradigm, to enhance the transferability of data quality ratings across various time series domains. Specifically, the goal is to learn an initialization of model parameters $\theta$ that enables rapid adaptation to a new time series domain with minimal additional fine-tuning efforts.

At each meta-training iteration, we sample a batch of tasks $\{\mathcal{T}_i\}_{i=1}^{B}$ from the task set, where each task corresponds to a specific dataset under a quality criterion (e.g., trend or frequency preference labels from a domain-specific subset of Time-300B). For each task $\mathcal{T}_i$, we randomly split its labeled examples into a support set $\mathcal{D}_i^{\text{support}}$ (used for adaptation) and a query set $\mathcal{D}_i^{\text{query}}$ (used for meta-updates).

For each task $\mathcal{T}_i$, TSRater aims to find task-specific parameters by minimizing the support loss:

$$\theta_i' = \arg\min_{\theta} \mathcal{L}_{\mathcal{T}_i}^{\text{support}}(\theta). \tag{5}$$

This is approximated by applying gradient descent steps using SignSGD:

$$\theta_i' = \theta - \alpha \cdot \text{sign}\left(\nabla_{\theta} \mathcal{L}_{\mathcal{T}_i}^{\text{support}}(\theta)\right). \tag{6}$$

---

**Algorithm 1** Meta-training TSRater with SignSGD

---

1: **Input:** Task set $\{\mathcal{T}_i\}_{i=1}^N$, inner learning rate $\alpha$, outer learning rate $\beta$
2: Initialize the TSRater model with parameters $\theta$
3: **repeat**
4:      Sample batch of tasks $\{\mathcal{T}_i\}_{i=1}^B$
5:      **for** all task $\mathcal{T}_i$ **do**
6:          Sample a support set $\mathcal{D}_i^{\text{support}}$ and a query set $\mathcal{D}_i^{\text{query}}$ from $\mathcal{T}_i$
7:          Compute inner-loss $\mathcal{L}_{\mathcal{T}_i}^{\text{support}}(\theta)$ using $\mathcal{D}_i^{\text{support}}$ and $\mathcal{L}_\theta$ in Eq.(3)
8:          Perform inner-loop adaptation: $\theta_i' \leftarrow \theta - \alpha \cdot \text{sign}\left(\nabla_\theta \mathcal{L}_{\mathcal{T}_i}^{\text{support}}(\theta)\right)$
9:      **end for**
10:     Compute meta-loss: $\mathcal{L}_{\text{meta}} \leftarrow \sum_{i=1}^B \mathcal{L}_{\mathcal{T}_i}^{\text{query}}(\theta_i')$ using each $\mathcal{D}_i^{\text{query}}$ and $\mathcal{L}_\theta$ in Eq.(3)
11:     Update meta-parameters: $\theta \leftarrow \theta - \beta \nabla_\theta \mathcal{L}_{\text{meta}}$
12: **until** convergence or max iterations reached
13: **Return:** Meta-trained parameters $\theta$

---

Although only one inner update is illustrated for simplicity, in practice, the model can perform one or more such updates during inner-loop training. This update uses the sign of the gradient direction instead of the exact gradient, which avoids expensive second-order derivative computations typically required in vanilla MAML (Finn et al., 2017). This makes the method more efficient and scalable in practice (Fan et al., 2021).

After adapting to each task, the updated parameters $\theta_i'$ are used to evaluate the query set and compute the query loss $\mathcal{L}_{\mathcal{T}_i}^{\text{query}}(\theta_i')$. The overall meta-objective is then formulated as minimizing the aggregated query losses across tasks:

$$\theta^* = \arg\min_\theta \mathcal{L}_{\text{meta}} = \arg\min_\theta \sum_{i=1}^B \mathcal{L}_{\mathcal{T}_i}^{\text{query}}(\theta_i'). \tag{7}$$

Finally, the outer-loop update is performed using standard SGD to minimize the meta-loss with respect to the original parameters $\theta$:

$$\theta = \theta - \beta \nabla_\theta \mathcal{L}_{\text{meta}} \tag{8}$$

Please note that the meta-optimization is performed over the initial model parameters $\theta$, while the meta-loss $\mathcal{L}_{\text{meta}}$ is evaluated using the adapted parameters $\theta_i'$. This meta-training loop is repeated until convergence or a predefined number of iterations is reached. The result is a TSRater model with parameters $\theta$ that can quickly adapt to new time series datasets using only a few labeled examples.

**First-Order Approximation via SignSGD.** To be self-contained, we follow (Fan et al., 2021) to present further details of deriving the SignSGD-based meta-learning training algorithm. In the standard MAML framework, the meta-gradient is computed by differentiating the query loss $\mathcal{L}_{\mathcal{T}_i}^{\text{query}}(\theta_i')$ with respect to the initial parameters $\theta$, through the adapted parameters $\theta_i'$:

$$\nabla_\theta \mathcal{L}_{\mathcal{T}_i}^{\text{query}}(\theta_i') = \frac{\partial \mathcal{L}_{\mathcal{T}_i}^{\text{query}}(\theta_i')}{\partial \theta_i'} \cdot \frac{\partial \theta_i'}{\partial \theta}. \tag{9}$$

When the inner-loop adaptation is performed using $M$-step vanilla gradient descent, the updates proceed recursively as:

$$\theta_i^{(0)} = \theta, \quad \theta_i^{(m+1)} = \theta_i^{(m)} - \alpha \nabla_{\theta_i^{(m)}} \mathcal{L}_{\mathcal{T}_i}^{\text{support}}(\theta_i^{(m)}), \quad m = 0, \ldots, M-1, \quad \theta_i' := \theta_i^{(M)}. \tag{10}$$

where $\theta_i' := \theta_i^{(M)}$ denotes the final adapted parameters after the inner loop.

Thus, the Jacobian $\frac{\partial \theta_i'}{\partial \theta}$ in Eq.(9) involves a chain of transformations across $M$ inner updates:

$$\frac{\partial \theta_i'}{\partial \theta} = \prod_{m=0}^{M-1} \left(I - \alpha \nabla_{\theta_i^{(m)}}^2 \mathcal{L}_{\mathcal{T}_i}^{\text{support}}(\theta_i^{(m)})\right). \tag{11}$$

It contains $M$ Hessian matrices (i.e., second-order derivatives), leading to significant computational and memory overhead during meta-training, particularly for high-dimensional models.

Instead of the vanilla gradient descent used in the inner loop, our approach employs **SignSGD** for parameter updates:

$$\theta_i^{(0)} = \theta, \quad \theta_i^{(m+1)} = \theta_i^{(m)} - \alpha \cdot \mathrm{sign}\left(\nabla_{\theta_i^{(m)}} \mathcal{L}_{\mathcal{T}_i}^{\mathrm{support}}(\theta_i^{(m)})\right), \quad m = 0, \ldots, M-1, \quad \theta_i' := \theta_i^{(M)}. \tag{12}$$

Each inner update now involves the sign operator, which is piecewise constant. Since

$$\frac{d}{dx}\mathrm{sign}(x) = \begin{cases} 0, & x \neq 0 \\ \mathrm{undefined}, & x = 0 \end{cases}, \tag{13}$$

we approximate the gradient propagation of each inner update as zero almost everywhere. As a result, the composite Jacobian simplifies to:

$$\frac{\partial \theta_i'}{\partial \theta} = \prod_{m=0}^{m-1} \left(I - \alpha \cdot \frac{\partial}{\partial \theta_i^{(m)}} \mathrm{sign}(\cdot)\right) = I, \tag{14}$$

yielding a **first-order approximation** of the meta-gradient:

$$\nabla_\theta \mathcal{L}_{\mathcal{T}_i}^{\mathrm{query}}(\theta_i') = \nabla_{\theta_i'} \mathcal{L}_{\mathcal{T}_i}^{\mathrm{query}}(\theta_i'). \tag{15}$$

This bypasses the need for computing second-order derivatives entirely, leading to a simplified first-order meta-gradient expression:

$$\nabla_\theta \mathcal{L}_{\mathrm{meta}} = \sum_{i=1}^{B} \nabla_{\theta_i'} \mathcal{L}_{\mathcal{T}_i}^{\mathrm{query}}(\theta_i'), \tag{16}$$

which aggregates the gradients from all tasks without backpropagating through the inner-loop updates. Consequently, SignSGD-based inner updates offer a highly efficient and scalable alternative to standard MAML, especially suited for large-scale meta-training.

### C.2 Details of Training Dataset Construction for Meta-learned TSRater

We construct 22 meta-learning tasks from the Time-300B corpus, covering a wide variety of real-world and synthetic time-series domains. Each task corresponds to a dataset labeled with pairwise quality judgments for one of the four quality criteria: trend, frequency, amplitude, or pattern. The selected datasets span nine major domains, as summarized in Table 8, ranging from energy, finance, and healthcare to transportation, web data, and synthetic sources.

To generate the pairwise quality judgments, we randomly sample and match pairs of time series within each dataset. Each time series is truncated to a maximum length of 128 to stay within the token limitations of LLMs. For every pair, we query GPT-4o-mini 20 times to assess which sequence exhibits higher quality under a specific criterion. Since we did not have access to the LLM's internal logits, we estimate the preference confidence based on the proportion of votes from the 20 generations. To reduce ambiguity, we prompt the LLM separately for each criterion, as querying for multiple criteria jointly was observed to degrade performance in our preliminary experiments.

In our experiments, we train and test the TSRater model using only the judgment samples $\{(\mathbf{B}_i, \mathbf{B}_j, p_{i \succ j})\}$ with a confidence greater than 50%. That means the confidence scores $p_{i \succ j}$ in these samples needs to satisfy $|2p_{i \succ j} - 1| \geq 0.5$, ensuring that we rely on more reliable judgments and filter out those with lower confidence. For each dataset, we collect 2000 valid preference judgments in total, consisting of 500 per criterion.

### C.3 Detailed Settings and Hyperparameter Selection of TSRater Model Training

We split each dataset into support, query, and test sets using a fixed ratio of 4:4:2. The support set is used for inner-loop adaptation in the meta-learning process, the query set is used for outer-loop meta-updates, and the test set serves as a held-out validation set to guide hyperparameter tuning.

Table 8: Domains and selected datasets comprising 22 meta-learning tasks for TSRater training.

| Domain | Selected Datasets |
|---|---|
| Energy | electricity_15min, electricity_weekly, solar_power |
| Finance | exchange_rate, tourism_monthly |
| Healthcare | hospital |
| Nature | weatherbench_daily, weatherbench_weekly, china_air_quality |
| Other | monash_m3_monthly, m1_monthly, m4_hourly, m4_quarterly, m4_weekly |
| Sales | m5, restaurant, favorita_sales |
| Synthetic | training_corpus_tsmixup_10m |
| Transport | monash_traffic, traffic_hourly, taxi_1h |
| Web | wiki_daily_100k |

To ensure the performance of our meta-learning framework, we conduct hyperparameter tuning using the Optuna library. The key hyperparameters considered for tuning include the meta-learning rate ($\beta$), the inner-loop learning rate ($\alpha$), the number of inner-loop adaptation steps ($M$), the number of meta-tasks per batch ($B_m$), the batch size for individual data samples within each task ($B_d$), and the number of meta-training epochs ($T$). Specifically, we search over the following ranges: $\beta \in [1 \times 10^{-4}, 5 \times 10^{-4}]$ (log-uniform), $\alpha \in [1 \times 10^{-3}, 5 \times 10^{-3}]$ (log-uniform), $M \in [10, 20]$, $B_m \in [5, 15]$, and $T \in [5, 10]$, while keeping $B_d$ fixed at 16. This tuning process allows us to identify a high-performance configuration for meta-learning.

After hyperparameter searching, the best-performing configuration is identified as follows: meta-learning rate $\beta = 2.5 \times 10^{-4}$, inner-loop learning rate $\alpha = 4 \times 10^{-3}$, inner-loop steps $M = 14$, meta-batch size $B_m = 10$, and training epochs $T = 7$. These values were used in all subsequent experiments.

### C.4 FURTHER EXPERIMENTAL RESULTS FOR META-LEARNED TSRATER

To evaluate the generalization ability of the meta-learned TSRater, we conducted experiments on a held-out dataset sampled from subsets of the healthcare domain (excluding the *hospital* dataset), which was not included in the 22 datasets used during meta-training. This held-out dataset consists of 2,000 samples in total, with 500 samples allocated for each evaluation criterion. This setup aims to assess the model's capability to adapt to unseen data distributions using limited data.

During evaluation, we employ a few-shot fine-tuning approach to adapt the meta-learned TSRater to the held-out dataset, using only 10 sample pairs for fine-tuning, while the remaining samples were reserved entirely for testing. The fine-tuning involves 10 adaptation steps with an adaptation learning rate of $1 \times 10^{-4}$. The evaluation is conducted through pairwise comparisons: for each pair of samples, TSRater generates scores and predicts their relative ranking. A prediction is deemed correct if the model's predicted preference matches the ground truth label. The overall evaluation accuracy is then calculated as the proportion of correctly predicted pairs.

To further demonstrate the advantage of meta-learning, we compare the performance of the meta-trained TSRater with several single-dataset raters, each trained independently on one specific dataset (from the benchmark datasets used in the main experiments; see Appendix D.1 for details). All models share the same architecture to ensure a fair comparison. The comparison is conducted on the held-out dataset introduced above, and focuses on the four key quality criteria. Table 9 reports the accuracy results, which show that the meta-trained TSRater consistently outperforms models trained on individual datasets across all four quality dimensions. These results validate the effectiveness of the meta-learning framework in acquiring transferable knowledge for time-series quality assessment in unseen domains.

**Additional Experiments for Analyzing Data Efficiency of Meta-Learned TSRater.** To further explore the data efficiency of the meta-learned TSRater, we conduct an additional experiment comparing its performance with the best-performing single-dataset raters identified in Table 9 for each of the four quality criteria. While the previous evaluation adopts a few-shot tuning setting with 10 sample pairs, we now vary the number of adaptation samples from 0 to 3, aiming to test the model's robustness under even more constrained supervision.

Table 9: Accuracy comparison on a held-out dataset between TSRater trained via meta-learning and single-dataset raters trained independently. "/" indicates that the model could not be trained for the corresponding criterion due to insufficient supervision. Bold indicates the best result; underline indicates the second best.

| Training Dataset (Method) | Trend | Frequency | Amplitude | Pattern |
|---|---|---|---|---|
| Time-300B (Meta-learning) | **0.7505** | **0.7515** | **0.7845** | **0.8121** |
| Traffic | 0.6424 | 0.6081 | / | 0.7262 |
| Weather | 0.7030 | 0.6484 | 0.7101 | 0.5990 |
| Electricity | 0.6293 | 0.4202 | 0.6091 | 0.6566 |
| Exchange Rate | 0.5848 | 0.5899 | 0.4646 | 0.5818 |
| M4-Daily | 0.6808 | 0.5545 | 0.3333 | 0.6879 |
| M4-Monthly | 0.4384 | 0.7111 | 0.5737 | 0.7737 |
| M4-Yearly | 0.7071 | 0.6717 | 0.6818 | 0.6697 |
| BME | 0.3566 | 0.4152 | / | 0.4091 |
| CBF | 0.3859 | 0.6101 | 0.3162 | 0.6465 |
| Handwriting | 0.6020 | 0.6505 | / | 0.7444 |
| MedicalImages | 0.6747 | 0.6545 | 0.4444 | 0.7475 |

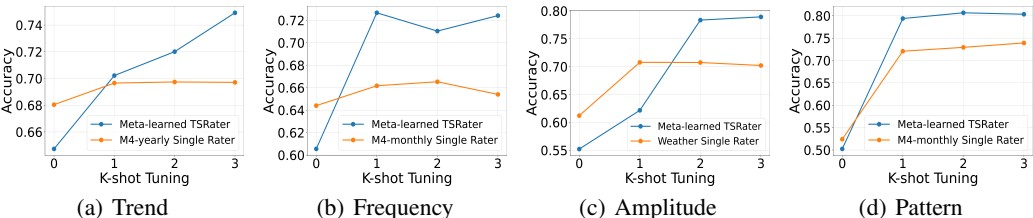

(a) Trend  (b) Frequency  (c) Amplitude  (d) Pattern

Figure 7: Few-shot adaptation results of the meta-learned TSRater compared with the best-performing single-dataset raters on the held-out healthcare dataset. Each subplot shows accuracy trends under 0 to 3 adaptation samples for one quality criterion.

As shown in Figure 7, the meta-learned TSRater exhibits strong adaptability to new tasks. Although it may not outperform the best single-dataset rater in the 0-shot or 1-shot setting for some criteria, its performance improves markedly with just a few more adaptation samples. With 2 or 3 samples, TSRater consistently achieves higher accuracy than any single-dataset model across all dimensions. This illustrates the strength of meta-learning in producing a model that is both easy to fine-tune and highly generalizable across unseen domains.

**Additional Experiments for Adaptability Evaluation on Eleven Benchmark Datasets.** Finally, we evaluate the meta-learned TSRater on the 11 benchmark datasets from the previous main experiments (see Appendix D.1 for details). For each dataset, 5% of labeled samples are used for fine-tuning, and the rest for testing. As shown in Table 10, TSRater achieves strong performance across all four quality criteria. The average accuracy is 0.8666 (trend), 0.8483 (frequency), 0.8843 (amplitude), and 0.8910 (pattern).

# D  ADDITIONAL DETAILS OF EXPERIMENT SETTINGS AND RESULTS

This section presents further experiment details and extended experiment results. The following contains three parts: (1) Section D.1 provides details of eleven benchmark datasets used in our main experiments; (1) Section D.2 provides details of baselines; (3) Section D.3 provides additional experiment results with five different time series models; (4) Section D.4 provides additional efficiency evaluation on benchmark datasets; (5) Section D.5 provides additional ablation studies on framework components; (6) Section D.6 provides extended evaluation on diverse experiment configurations; (7) Section D.7 provides additional finetuning results measured by the MAE evaluation metric; (8) Section D.8 provides additional experiment results of multi-LLM ensemble judgments.

Table 10: Evaluation of the meta-learned TSRater on 11 benchmark datasets. Each model is fine-tuned using 5% of the labeled data and tested on the remaining samples. Slash (/) indicates that the corresponding criterion is not applicable to the dataset.

| Datasets | Trend | Frequency | Amplitude | Pattern |
|---|---|---|---|---|
| Traffic | 0.8953 | 0.8267 | / | 0.9202 |
| Weather | 0.8016 | 0.8315 | 0.9095 | 0.8742 |
| Electricity | 0.8924 | 0.8706 | 0.9700 | 0.8771 |
| Exchange rate | 0.8932 | 0.8679 | 0.9200 | 0.9238 |
| m4-daily | 0.7958 | 0.8357 | 0.7574 | 0.8216 |
| m4-monthly | 0.7772 | 0.7494 | 0.8363 | 0.7834 |
| m4-yearly | 0.7621 | 0.7691 | 0.7457 | 0.7867 |
| BME | 0.8333 | 0.8461 | / | 1.0000 |
| CBF | 1.0000 | 1.0000 | 1.0000 | 1.0000 |
| Handwriting | 0.9333 | 0.8491 | / | 0.8667 |
| MedicalImages | 0.9481 | 0.8847 | 0.9353 | 0.9474 |
| **Average** | **0.8666** | **0.8483** | **0.8843** | **0.8910** |

## D.1 DETAILS OF ELEVEN BENCHMARK DATASETS

This section provides detailed descriptions of the eleven benchmark datasets used in our experiments in Section 4, summarized in Table 11. Below is a more detailed overview of each dataset (Li et al., 2025).

- Electricity (Trindade, 2015): It contains hourly electricity consumption (Kwh) data of 321 clients from 2012 to 2014. We set 'MT_320' as the target value in the univariate setting.

- Exchange Rate (Lai et al., 2018): It records daily currency exchange rates for eight countries, covering the period from 1990 to 2016. We set 'Singapore' as the target value.

- Traffic: It consists of hourly occupancy rates of freeway lanes across the San Francisco Bay Area from 2015 to 2016. We set 'Sensor_861' as the target value.

- Weather: It includes hourly meteorological data (temperature, humidity, wind speed, etc.) from nearly 1600 locations in the US. We set 'wet_bulb' as the target value.

- M4 (Makridakis et al., 2018): The M4 dataset is a collection of 100,000 time series used for the fourth edition of the Makridakis forecasting Competition. It consists of time series of yearly, quarterly, monthly, and other (weekly, daily, and hourly) data. We utilize its **Yearly**, **Monthly**, and **Daily** subsets, which span different time frequencies and forecast horizons.

- MedicalImages: A univariate dataset consisting over 1000 time series samples, each representing pixel intensity variations over 99 time steps, with 10 distinct classes.

- CBF: A synthetic univariate dataset with 930 samples, each of which includes 128 time steps representing distinct patterns generated by geometric shapes, with 3 classes.

- BME: A generated univariate dataset containing 180 samples categorized in 3 classes, with each sample being a time series of length 128.

- Handwriting: A multivariate dataset with 1000 samples capturing temporal pen movements during handwriting tasks. All samples have 3 channels and 152 time steps, categorized in 26 classes.

## D.2 DETAILS OF BASELINES

**DataShapley** (Ghorbani & Zou, 2019): This method calculates the contribution of each data point to the model's performance based on Shapley values.

**KNNShapley** (Jia et al., 2019a): As a variation of DataShapley, KNNShapley uses k-nearest neighbors algorithms to estimate the contribution of data points.

Table 11: Details of the eleven benchmark datasets used in our experiments.

| Task | Dataset | Channels | Series Length | Data Size | Information |
|---|---|---|---|---|---|
| Long-term forecasting | Electricity | 321 | - | 25507 | Electricity (Hourly) |
| | Traffic | 862 | - | 16747 | Transportation (Hourly) |
| | Weather | 21 | - | 51899 | Weather (10 mins) |
| | Exchange | 8 | - | 6791 | Exchange rate(Daily) |
| Short-term forecasting | M4-Yearly | 1 | - | 23000 | - |
| | M4-Monthly | 1 | - | 48000 | - |
| | M4-Daily | 1 | - | 24000 | - |
| Classification | MedicalImages | 1 | 99 | 1141 | Pixel Intensity (10 classes) |
| | CBF | 1 | 128 | 930 | Geometric Patterns (3 classes) |
| | BME | 1 | 128 | 180 | Synthetic Signals (3 classes) |
| | Handwriting | 3 | 152 | 1000 | Pen Trajectory (26 classes) |

Table 12: Long-term forecasting (RMSE) results of different data selection methods across representative models and datasets. The best results are **bolded**, and the second-best results are underlined.

| Model | Selection method | Electricity | Exchange Rate | Traffic | Weather |
|---|---|---|---|---|---|
| iTransformer | Random | 0.342 | 0.267 | 0.288 | 0.509 |
| | DataOob | 0.309 | 0.246 | 0.275 | 0.569 |
| | DataShapley | 0.316 | 0.272 | 0.285 | 0.452 |
| | KNNShapley | 0.321 | 0.280 | 0.276 | 0.489 |
| | TimeInf | 0.307 | 0.249 | 0.277 | 0.454 |
| | TSRating | **0.300** | **0.227** | **0.270** | **0.444** |
| TimeMixer | Random | 0.406 | 0.235 | 0.311 | 0.537 |
| | DataOob | 0.416 | 0.226 | 0.317 | 0.496 |
| | DataShapley | 0.396 | 0.238 | 0.302 | 0.466 |
| | KNNShapley | 0.434 | 0.316 | 0.285 | 0.517 |
| | TimeInf | 0.404 | 0.224 | 0.308 | 0.461 |
| | TSRating | **0.345** | **0.222** | **0.282** | **0.433** |

**DataOob** (Kwon & Zou, 2023): DataOob uses out-of-bag (OOB) estimates to assess the contribution of each data point based on its impact on model performance.

**TimeInf** (Zhang et al., 2024b): TimeInf uses influence functions to attribute model predictions to individual time points while preserving temporal structures.

**Random**: We also randomly select time series data as a control group to assess the performance of sophisticated data evaluation methods against an arbitrary selection of data points.

Table 13: Short-term forecasting (MAPE) results of different data selection methods across representative models and datasets. Best results are **bolded**, second-best are underlined.

| Model | Selection Method | M4_Yearly | M4_Monthly | M4_Daily |
|---|---|---|---|---|
| Nonstationary Transformer | Random | 3.220 | 1.799 | **1.572** |
| | DataOob | 2.676 | 1.960 | 1.731 |
| | DataShapley | 3.571 | 1.765 | 1.689 |
| | KNNShapley | 3.840 | 1.780 | 1.751 |
| | TimeInf | 2.961 | 1.673 | 1.720 |
| | TSRating | **2.657** | **1.605** | 1.582 |
| DLinear | Random | 3.856 | 1.348 | 1.413 |
| | DataOob | 3.185 | 1.323 | 1.548 |
| | DataShapley | **2.234** | 1.326 | 1.480 |
| | KNNShapley | 3.334 | 1.313 | **1.389** |
| | TimeInf | 3.053 | 1.380 | 1.505 |
| | TSRating | 2.764 | **1.312** | 1.411 |

Table 14: Classification accuracy of different data selection methods across representative models and datasets. Best results are **bolded**, second-best are underlined.

| Model | Selection Method | MedicalImages | CBF | BME | Handwriting |
|---|---|---|---|---|---|
| Informer | Random | 0.599 | 0.617 | 0.580 | 0.193 |
| | DataOob | **0.626** | 0.622 | 0.467 | 0.196 |
| | DataShapley | 0.624 | 0.603 | 0.533 | 0.187 |
| | KNNShapley | 0.621 | **0.650** | 0.593 | 0.186 |
| | TimeInf | 0.614 | 0.613 | 0.507 | 0.195 |
| | TSRating | 0.625 | 0.648 | **0.633** | **0.209** |
| Nonstationary Transformer | Random | 0.567 | 0.521 | 0.604 | 0.207 |
| | DataOob | 0.611 | 0.697 | 0.565 | **0.224** |
| | DataShapley | **0.638** | 0.729 | 0.580 | 0.179 |
| | KNNShapley | 0.609 | 0.533 | 0.563 | 0.188 |
| | TimeInf | 0.597 | 0.662 | 0.580 | 0.205 |
| | TSRating | 0.628 | **0.733** | **0.612** | 0.213 |

## D.3 EXTENDED EXPERIMENT RESULTS ON FIVE DIFFERENT TIME SERIES MODELS

To complement the main results presented in Section 4.1, we report additional experiments using two competitive models for each task category to further validate the generalizability and robustness of TSRating across diverse model architectures.

Specifically, we select iTransformer (Liu et al., 2023) and TimeMixer (Wang et al., 2024b) for long-term forecasting, Nonstationary Transformer (Liu et al., 2022) and DLinear (Zeng et al., 2023a) for short-term forecasting, and Informer and Nonstationary Transformer for classification tasks. These models are highly competitive in their respective domains, representing recent state-of-the-art architectures and offering a broader evaluation perspective.

Table 12 shows the RMSE performance on long-term forecasting benchmarks, where TSRating consistently achieves the best performance across all datasets. This highlights its ability to enhance models that capture long-range temporal dependencies. In Table 13, we report MAPE results on M4 subsets for short-term forecasting. TSRating again demonstrates strong performance, often achieving the best or second-best results, which underscores its adaptability to different forecasting horizons and finer-grained temporal patterns. Finally, Table 14 presents classification accuracy, where TSRating performs robustly across datasets with varied structures and difficulty, confirming its generalizability across both regression and classification settings.

These results corroborate the findings in the main text and reinforce that TSRating provides robust and domain-agnostic data selection benefits across different time series learning tasks.

Table 15: Efficiency of different selection methods across Long-term forecasting datasets (in seconds).

| Selection Method | Electricity | Traffic | Weather | Exchange Rate |
|---|---|---|---|---|
| DataOob | 20.82 | 22.56 | 20.56 | 21.17 |
| DataShapley | 1928.71 | 350.60 | 897.57 | 738.16 |
| KNNShapley | 0.45 | 0.63 | 0.47 | 0.43 |
| TimeInf | 19.12 | 26.00 | 26.74 | 19.08 |
| TSRating (amortized) | 6.01 | 3.42 | 5.05 | 4.18 |
| TSRating | 1.23 | 0.99 | 1.20 | 1.35 |

## D.4 ADDITIONAL EFFICIENCY EVALUATION ON BENCHMARK DATASETS

To complement the main efficiency analysis in Section 4.1, we further evaluate the efficiency of TSRating on benchmark datasets commonly used for time series prediction tasks. As shown in Table 15, the proposed TSRating demonstrates superior efficiency compared to all baseline methods except KNNShapley. While KNNShapley is faster due to its simpler model and lower computational complexity, it sacrifices evaluation accuracy. Note that we only measure the inference time for all selection methods, without additional preprocessing time, e.g., the TSRater training time. The

amortized time for TSRating, as shown in Table 15, reflects the time spent on LLM judgment and TSRater training, which is divided across the total number of ratings. This amortized time provides a more balanced view of the overall time required for our TSRating.

## D.5 ABLATION STUDIES ON FRAMEWORK COMPONENTS

To further clarify the contributions of each component in TSRating, we conducted extended ablation experiments. While Section 4.3 already reported ablations on quality criteria and LLM variants, here we provide additional analyses on the encoder choice, and the supervision mechanism from LLMs.

**Effect of the MOMENT Encoder.** We replaced the MOMENT encoder (109M parameters) with a lightweight Transformer-based encoder (10% of the size) and evaluated TSRater on the Electricity dataset. As shown in Table 16, TSRater's performance deteriorated across all forecasting models, e.g., RMSE for Linear increased from 1.390 to 1.541, and for PatchTST from 0.397 to 0.419. This indicates that the pretrained temporal representations from MOMENT are crucial for robust rating performance.

Table 16: Ablation on the MOMENT encoder using the Weather dataset. Best results are **bolded**.

| Method | Linear | CNN | PatchTST | iTransformer | TimeMixer |
|---|---|---|---|---|---|
| TSRating with MOMENT | **1.390** | **1.511** | **0.397** | **0.300** | **0.345** |
| w/o MOMENT | 1.541 | 1.534 | 0.419 | 0.307 | 0.366 |

**Pairwise vs. Direct LLM Supervision.** TSRating relies on LLMs to provide qualitative, comparative judgments under weak supervision. To validate this design choice, we constructed synthetic datasets with ground-truth labels for each quality criterion (see Appendix B.2 for more details) and compared the accuracy of LLM pairwise judgments against direct numerical scoring. As shown in Table 17, pairwise judgments consistently achieve higher accuracy across all four criteria, supporting our use of comparison-based signals instead of direct scoring.

Table 17: Accuracy of LLM supervision strategies on synthetic datasets with ground-truth quality criteria.

| Criterion | LLM (Pairwise) | LLM (Direct Score) |
|---|---|---|
| Trend | **0.995** | 0.831 |
| Frequency | **1.000** | 0.862 |
| Amplitude | **0.998** | 0.871 |
| Pattern | **0.968** | 0.834 |

Taken together, these extended ablations confirm that TSRating's performance benefits from pretrained temporal encoders, and pairwise LLM supervision. Each component contributes to the robustness and generalization ability of the framework, validating the design choices made in this work.

## D.6 EXTENDED EVALUATION ON DIVERSE EXPERIMENT CONFIGURATIONS

In practical applications, data quality evaluation often requires design choices related to sample selection, segmentation, and forecasting configuration. TSRating introduces two task-level parameters: the **selection ratio**, which controls the fraction of retained samples after scoring, and the **block size**, which defines the temporal span of each segment. While the defaults (0.5 for selection ratio and 128 for long-term forecasting block size) follow common practice for fair comparison with prior works, it is essential to assess whether TSRating remains effective under alternative configurations. To this end, we conduct additional experiments on the Traffic dataset with the iTransformer backbone, systematically varying both parameters. Beyond this, we further evaluate TSRating's robustness under **longer forecast horizons**, where prediction errors typically accumulate, and extend the framework to **multivariate-to-multivariate (M2M) forecasting**, validating its applicability beyond the univariate-to-univariate (U2U) setup used in the main experiments. Together, these studies provide a more comprehensive view of TSRating's generalization ability across diverse experimental settings.

**Selection Ratio Experiments.** The impact of selection ratio on TSRating's performance is shown in Table 18. TSRating consistently outperforms other methods at nearly all tested selection ratios, indicating that the framework is robust to variations in the fraction of retained samples. It further demonstrate TSRating's generalization capability and reliability regardless of how many samples are selected.

Table 18: Forecasting performance (RMSE) under different selection ratios on the Traffic dataset with iTransformer. Best results are **bolded**, second-best are underlined.

| Method | 0.2 | 0.4 | 0.6 | 0.8 | 1.0 |
|---|---|---|---|---|---|
| Random | 0.382 | 0.366 | 0.356 | 0.352 | 0.339 |
| DataOob | **0.371** | 0.364 | 0.354 | 0.346 | 0.339 |
| DataShapley | 0.381 | **0.356** | 0.353 | 0.343 | 0.339 |
| KNNShapley | 0.378 | 0.363 | 0.355 | 0.346 | 0.339 |
| TimeInf | 0.377 | 0.360 | 0.357 | 0.348 | 0.339 |
| TSRating | 0.372 | **0.356** | **0.349** | **0.342** | 0.339 |

**Block Size Experiments.** In addition, we also explored the effect of block size on TSRating's performance. As shown in Table 19, TSRating maintains its advantage over other methods even when the block size varies. Specifically, TSRating achieves the lowest RMSE when compared to other methods, irrespective of whether smaller or larger block sizes are used. These results further demonstrate the robustness of TSRating to changes in block size, confirming its generalizability across different segment lengths.

Table 19: Forecasting performance (RMSE) under different block sizes on the Traffic dataset with iTransformer. Best results are **bolded**, second-best are underlined.

| Method | 128 | 288 | 432 |
|---|---|---|---|
| Random | 0.347 | 0.349 | 0.351 |
| DataOob | 0.341 | 0.347 | 0.350 |
| DataShapley | **0.338** | 0.345 | 0.347 |
| KNNShapley | 0.341 | 0.347 | 0.348 |
| TimeInf | 0.341 | 0.346 | **0.342** |
| TSRating | 0.339 | **0.341** | **0.342** |

**Forecast Horizons.** We further examined TSRating under different forecast horizons using the Traffic dataset with the PatchTST backbone. Table 20 reports the mean RMSE $\pm$ standard deviation over five random seeds for horizons of 192, 336, and 720 (prediction length fixed at 96). TSRating consistently performs on par with or better than competing methods across all horizons. Importantly, at longer horizons (336 and 720), where forecast errors typically accumulate, TSRating continues to deliver robust and stable performance. These results underscore TSRating's ability to generalize under increasingly challenging forecasting conditions.

Table 20: Forecasting performance (RMSE $\pm$ std.) under different horizons on the Traffic dataset with PatchTST. Best results are **bolded**, second-best are underlined.

| Method | 192 | 336 | 720 |
|---|---|---|---|
| Random | $0.347 \pm 0.0021$ | $0.352 \pm 0.0019$ | $0.355 \pm 0.0025$ |
| DataOob | $0.342 \pm 0.0017$ | $0.348 \pm 0.0022$ | $0.355 \pm 0.0018$ |
| DataShapley | $\mathbf{0.337} \pm 0.0020$ | $0.346 \pm 0.0018$ | $0.357 \pm 0.0026$ |
| KNNShapley | $0.341 \pm 0.0019$ | $0.348 \pm 0.0017$ | $0.358 \pm 0.0023$ |
| TimeInf | $0.344 \pm 0.0016$ | $\underline{0.345} \pm 0.0015$ | $\underline{0.347} \pm 0.0019$ |
| TSRating | $\underline{0.339} \pm 0.0015$ | $\mathbf{0.342} \pm 0.0013$ | $\mathbf{0.346} \pm 0.0020$ |

**Extension to Multivariate Forecasting (M2M).** In our main experiments, the forecasting task is conducted under the U2U (univariate-to-univariate) configuration. TSRating, however, is naturally extensible to M2M (multivariate-to-multivariate) forecasting, as described in Section 3.2. For

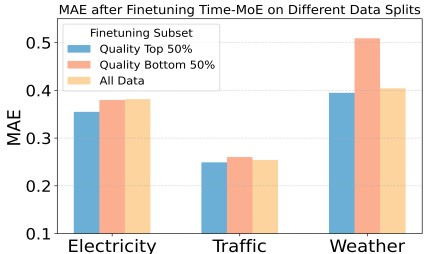 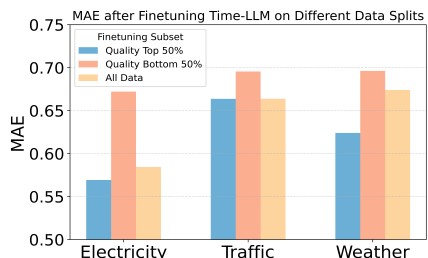

Figure 8: Mean Absolute Error (MAE) on test sets after finetuning time series foundation models (**Left:** Time-MoE, **Right:** Time-LLM) using different subsets of training data across three datasets. Each model is fine-tuned with either the top 50% highest-quality data, the bottom 50%, or the full training set. Finetuning on higher-quality subsets consistently leads to lower MAEs, underscoring the effectiveness of TSRating in guiding data selection.

multivariate time series, we compute quality scores by aggregating LLM pairwise judgments across all channels (Qiu et al., 2025a). To validate TSRating under the M2M configuration, we conducted experiments on the Electricity dataset using three representative forecasting backbones (Linear, CNN, and PatchTST). Results in Table 21 indicate that TSRating not only supports U2U forecasting but also extends seamlessly to M2M settings, consistently achieving superior performance across different forecasting architectures.

Table 21: Forecasting performance (RMSE) under the M2M configuration on the Electricity dataset. TSRating maintains consistent advantages across different forecasting backbones. Best results are **bolded**, second-best are underlined.

| Method | Linear | CNN | PatchTST |
|---|---|---|---|
| Random | 2.257 | 1.943 | 0.856 |
| DataOob | 2.014 | 1.919 | 0.854 |
| DataShapley | 2.048 | 1.917 | 0.854 |
| KNNShapley | 2.090 | 1.922 | 0.854 |
| TimeInf | 2.061 | 1.921 | 0.853 |
| TSRating | **2.013** | **1.913** | **0.837** |

The results across all these experimental settings—selection ratio, block size, and forecasting configurations, demonstrate that TSRating consistently achieves strong performance. Its effectiveness is largely unaffected by variations in parameter choices or forecasting setups, underscoring the framework's robustness and adaptability. These findings highlight TSRating as a practical and generalizable solution for time series data quality evaluation across diverse conditions.

### D.7 ADDITIONAL FINETUNING EXPERIMENT RESULTS WITH MAE METRIC

As an additional evaluation of our finetuning experiment on time series foundation models, we report the Mean Absolute Error (MAE) results to complement the MSE findings in the main text presented in Section 4.4. As shown in Figure 8, both Time-MoE and Time-LLM also achieve noticeably lower MAE values when fine-tuned on the top 50% high-quality data selected by TSRating, compared to the bottom 50%. This trend is consistent across all datasets evaluated, further confirming that our data quality ratings are robust across different error metrics. These results reinforce the practical utility of TSRating in enhancing downstream performance through effective data curation.

### D.8 ASSESSING THE EFFECTIVENESS OF MULTI-LLM ENSEMBLE JUDGMENTS

We further investigate a multi-LLM ensemble strategy to improve the robustness of LLM-derived judgments. In this experiment on the Electricity dataset, TSRating aggregates criterion-level judgments from three different LLMs (GPT-4o-mini, Gemini-2.0, and Claude-3.5). For each sample pair, the final preference label is determined through a majority voting rule across models. The

aggregated judgments are subsequently used for data selection prior to downstream model training. The forecasting results are reported in Table 22.

Overall, the ensemble approach provides comparable or slightly improved forecasting performance relative to using a single LLM. These results suggest that model diversity can enhance judgment stability, reducing idiosyncratic biases from individual foundation models. Such ensemble strategies represent a complementary direction for improving the reliability of LLM-driven time-series quality assessment.

Table 22: Forecasting RMSE using individual LLMs vs. multi-LLM ensemble judgments on the Electricity dataset.

| Model | Linear | CNN | PatchTST | iTransformer | TimeMixer |
|---|---|---|---|---|---|
| GPT-4o-mini | 1.390 | 1.511 | 0.397 | 0.300 | 0.345 |
| Gemini-2.0 | 1.410 | 1.501 | 0.402 | 0.304 | 0.323 |
| Claude-3.5 | 1.406 | 1.508 | 0.404 | 0.307 | 0.334 |
| Ensemble | 1.391 | 1.505 | 0.389 | 0.303 | 0.333 |

## E  ADAPTIVE AGGREGATION OF QUALITY CRITERIA

The current TSRating framework combines multiple quality criteria using simple averaging. While straightforward and effective, adaptive aggregation strategies, such as weighted averaging or attention mechanisms, may offer further improvements by emphasizing more informative criteria depending on the data domain or task.

To investigate this, we conduct an experiment on the Weather dataset using a weighted average scheme where criterion weights are assigned based on domain expertise: Trend (0.30), Frequency (0.10), Amplitude (0.25), and Pattern (0.35). These weights reflect the understanding that trend and pattern capture key structural behaviors, amplitude reflects the magnitude of changes, and frequency receives a lower weight due to noise sensitivity and redundancy with other criteria. Table 23 reports the forecasting RMSE results on the top 50% of selected samples across five downstream models.

Table 23: Forecasting RMSE with simple average versus weighted average aggregation of criteria on the Weather dataset.

| Aggregation | Linear | CNN | PatchTST | iTransformer | TimeMixer |
|---|---|---|---|---|---|
| Simple average | 0.611 | 0.734 | 0.467 | 0.444 | 0.433 |
| Weighted average | 0.611 | 0.719 | 0.473 | 0.456 | 0.424 |

The results indicate that both simple and weighted averaging achieve comparable performance, with no consistent advantage observed for weighted aggregation in this setting. This suggests that simple averaging already provides a robust and effective strategy for combining criterion scores.

Looking forward, we plan to explore more flexible and adaptive aggregation mechanisms. One promising direction is to learn criterion weights dynamically during meta-rater training. Specifically, the meta-rater could predict scores through a weighted combination of criterion-specific features, where the weights are initialized uniformly but updated via backpropagation to minimize validation loss across diverse tasks and domains. This approach would enable the model to automatically adjust criterion importance in response to varying domain characteristics, leading to potentially more generalizable rating functions. Although such extensions could further enhance performance, they do not diminish the novelty or significance of the current TSRating framework and its demonstrated capabilities.

## F  EXTENDED CASE STUDY ON ANOMALY-AWARE DATA SELECTION

To further demonstrate the extensibility of our proposed framework, we include an additional study in which TSRating is adapted to support task-specific criteria beyond the four general-quality metrics

Table 24: F1-score performance on the MSL anomaly detection task when using different data selection methods for training. TSRating denotes the extended version incorporating the anomaly-aware criterion.

| Method | Linear | CNN | PatchTST | iTransformer | TimeMixer |
|---|---|---|---|---|---|
| Random | 0.7678 | 0.7953 | 0.7936 | 0.7912 | 0.7897 |
| DataShapley | 0.7780 | 0.8144 | 0.8140 | 0.8125 | 0.8103 |
| TimeInf | 0.7909 | 0.8159 | 0.8140 | 0.8137 | 0.8120 |
| **TSRating** | **0.8253** | **0.8480** | **0.8506** | **0.8498** | **0.8485** |

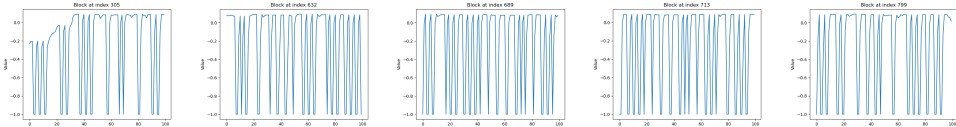

(a) Anomalous samples selected by TSRating with an task-specific anomaly-emphasized criterion.

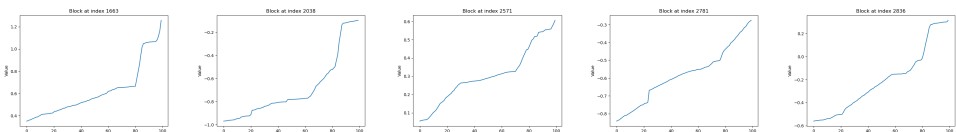

(b) Normal samples selected by TSRating with the original four criteria.

Figure 9: Visualization of samples selected on the MSL dataset for anomaly detection task, showing that TSRating can adapt to task-specific criteria that highlight irregularities, successfully preserving meaningful anomalous sequences while still maintaining reliable identification of normal data.

used in the main paper. While the original design primarily focuses on selecting high-quality normal sequences, certain downstream scenarios, such as anomaly detection, require explicitly preserving irregular but semantically important temporal behaviors. Motivated by this, we extend TSRating by incorporating an auxiliary criterion termed *Anomaly*, which aims to assess irregular or anomalous fluctuations in a sequence that may be crucial for accurate data selection.

We evaluate this enhanced framework on the MSL dataset under a binary anomaly detection setting (Darban et al., 2025b;a). TSRating first assesses each sequence combining the original four criteria to identify reliable normal samples. It then applies the anomaly criterion to capture irregular but important temporal behaviors. Through this two-fold scoring mechanism, the method preserves both reliable normal samples and anomalous but meaningful samples. The selected subsets are used to train five representative downstream models: Linear, CNN, PatchTST, iTransformer, and TimeMixer (Qiu et al., 2025b). We compare against Random selection as well as two established data valuation methods: DataShapley and TimeInf (Zhang et al., 2024b). Performance is reported in terms of F1-score, as summarized in Table 24. Beyond quantitative results, we further visualize the samples selected under the original four criteria and the anomaly criterion to provide intuitive insights into the behavior of the proposed anomaly-aware TSRating, as shown in Figure 9.

These results illustrate that incorporating anomaly-aware signals enables TSRating to better preserve meaningful irregular patterns that are easily overlooked by general-quality assessment schemes. This case study highlights the flexibility of TSRating to integrate additional criteria tailored to specific downstream requirements, thereby enhancing its applicability to diverse real-world time-series scenarios.

# G  FURTHER EVALUATION OF LLM JUDGMENT CONSISTENCY AND ROBUSTNESS

While leveraging large language models (LLMs) for time-series data quality evaluation presents a promising avenue, their judgments inherently face challenges such as limited prior validation, lack of consensus analysis among different LLMs, absence of human expert adjudication, and insufficient

quantification of uncertainty. Moreover, the stability and consistency of pairwise comparisons across varying prompt designs, parameter configurations, and model architectures remain underexplored. To rigorously assess these factors, we conduct comprehensive experiments aimed at characterizing the robustness, reliability, and agreement of LLM-based judgments within the TSRating framework.

**Synthetic Dataset Evaluation.** We extend beyond our initial small-scale validation by constructing a larger synthetic dataset designed to isolate and control the four key quality criteria: trend, frequency, amplitude, and pattern. This dataset is generated by systematically combining base signal templates (e.g., sine waves, linear trends, autoregressive processes) with controlled noise and anomaly injections, following a scalable pipeline inspired by prior literature (Cai et al., 2024). We collect 4,000 pairwise comparisons with well-defined ground-truth differences and evaluate four representative LLMs: GPT-4o-mini, Gemini-2.0, Claude-3.5, and DeepSeek-V3. As shown in Table 25, all models demonstrate high accuracy in recognizing these interpretable structures under idealized conditions.

Table 25: Accuracy of LLM judgments on synthetic dataset across four quality criteria.

| Criterion | GPT-4o-mini | Gemini-2.0 | Claude-3.5 | DeepSeek-V3 |
|---|---|---|---|---|
| Trend | 0.953 | 0.958 | 0.989 | 0.956 |
| Frequency | 0.967 | 0.924 | 0.942 | 0.997 |
| Amplitude | 0.985 | 0.992 | 0.964 | 0.972 |
| Pattern | 0.971 | 0.996 | 0.977 | 0.970 |

**Real-World Dataset and Human Alignment.** To validate the applicability of LLM judgments to practical scenarios, we evaluate agreement with human expert annotations on the Electricity dataset. An independent expert with domain experience, uninvolved in method development, labeled sample pairs according to the four quality criteria with intuitive qualitative guidelines (e.g., clarity of long-term trends, periodic consistency, meaningful amplitude variations, recognizable temporal patterns). Table 26 reports high LLM–human agreement rates, mostly exceeding 87%, indicating that TSRating's scoring aligns well with human understanding of time-series quality in real-world contexts.

Table 26: Agreement between LLM judgments and human expert labels on the Electricity dataset.

| Criterion | GPT-4o-mini | Gemini-2.0 | Claude-3.5 |
|---|---|---|---|
| Trend | 0.9733 | 0.9254 | 0.8844 |
| Frequency | 0.9367 | 0.8984 | 0.8911 |
| Amplitude | 1.0000 | 0.9000 | 0.9242 |
| Pattern | 0.9041 | 0.8811 | 0.8736 |

**Inter-LLM Agreement.** We assess the consistency of pairwise judgments across different LLM pairs on the same Electricity dataset. Table 27 shows strong agreement among GPT-4o-mini, Gemini-2.0, and Claude-3.5, further supporting the reliability of the rating process across model variants.

Table 27: Inter-LLM agreement rates on the Electricity dataset.

| Criterion | GPT-4o-mini vs Gemini-2.0 | GPT-4o-mini vs Claude-3.5 | Gemini-2.0 vs Claude-3.5 |
|---|---|---|---|
| Trend | 0.9259 | 0.9133 | 0.9065 |
| Frequency | 0.8917 | 0.9657 | 0.9000 |
| Amplitude | 1.0000 | 1.0000 | 0.9091 |
| Pattern | 0.9531 | 0.8722 | 0.9038 |

**Prompt Sensitivity.** We evaluate the effect of prompt variations by swapping the order of examples and instructions, and by paraphrasing prompt descriptions. The results, summarized in Tables 28 and 29, show very high agreement, indicating limited impact of prompt wording or structure on judgment consistency.

**Bradley-Terry Rating Robustness.** We test the robustness of the Bradley-Terry rating method by injecting controlled amounts of noise into the synthetic pairwise labels and measuring Spearman

Table 28: Agreement between original and order-swapped prompts across quality criteria.

| Quality Criterion | Agreement |
|---|---|
| Trend | 1.0000 |
| Frequency | 1.0000 |
| Amplitude | 1.0000 |
| Pattern | 1.0000 |

Table 29: Agreement between original and paraphrased prompts across quality criteria.

| Quality Criterion | Agreement |
|---|---|
| Trend | 0.9888 |
| Frequency | 1.0000 |
| Amplitude | 1.0000 |
| Pattern | 1.0000 |

correlation between noisy and original rankings. Table 30 shows the ranking remains stable with noise levels up to 8%, which is within a controllable range relative to the inherent inaccuracy of LLM judgments.

Table 30: Spearman correlation of Bradley-Terry ratings under increasing label noise.

| Noise Level | 0% | 2% | 4% | 6% | 8% |
|---|---|---|---|---|---|
| Correlation | 1.000 | 0.9854 | 0.9715 | 0.9496 | 0.9257 |

**LLM Judgment Uncertainty.** We analyze the effect of sampling randomness by varying the temperature parameter (0.1, 0.5, 1.0) of GPT-4o-mini on the Trend criterion judgments. Results in Table 31 reveal strong agreement across settings, indicating low uncertainty impact within the tested parameter ranges.

Table 31: Agreement between GPT-4o-mini judgments across different decoding temperature settings.

| Comparison | Agreement |
|---|---|
| 0.1 vs 0.5 | 1.000 |
| 0.1 vs 1.0 | 1.000 |
| 0.5 vs 1.0 | 1.000 |

These additional experiments demonstrate that, despite inherent limitations and potential biases, LLM judgments exhibit sufficient consistency and robustness across diverse prompts, model variants, noise levels, and human alignment. While future work may incorporate uncertainty calibration, enhanced prompt design, and expert adjudication, our results validate LLM-based evaluation as a promising and practical foundation for time-series rating.

# H  LIMITATIONS

While TSRating has demonstrated strong performance in assessing time series quality, it also presents several limitations that point to promising directions for future work.

One limitation of TSRating is the use of a uniform average for aggregating different criteria. While this approach has shown high accuracy in quality estimation, we could further explore non-uniform averaging methods or dynamic weighting schemes to better account for the differing importance of each criterion, potentially enhancing the model's performance.

Another limitation lies in the uniform intervals for block segmentation. The current method assumes a consistent change in the time series data across timestamps, but this may not always reflect the

underlying temporal dynamics. Future work could explore dynamic segmentation strategies, adapting the intervals based on the data's changing frequency or other temporal characteristics.

## I    THE USE OF LARGE LANGUAGE MODELS

In this work, large language models (LLMs) were employed solely as a general-purpose tool to improve clarity, coherence, and readability of the manuscript text. The LLMs did not participate in formulating research ideas, designing experiments, or analyzing results. All scientific content, experimental procedures, data processing, and interpretations were independently conducted by the authors.

While LLMs assisted with language refinement, the authors remain fully responsible for all content, including any portions revised with LLM support. No LLM was considered an author or formal contributor beyond its role in writing assistance.

