# OpenReview forum: "Rating Quality of Diverse Time Series Data by Meta-learning from LLM Judgment"
_ICLR.cc/2026/Conference — ICLR 2026 Poster_

### Official Review · Reviewer_pUfp · 2025-10-26

**Soundness:** 3
**Presentation:** 3
**Contribution:** 4
**Rating:** 6
**Confidence:** 4

**Summary:**

This paper proposes TSRating, a novel framework for evaluating the quality of diverse time series data, addressing limitations in existing methods regarding cross domain generalization and efficiency. The core idea involves using LLMs guided by specific prompts based on TS characteristics:trend, frequency, amplitude, pattern, to make pairwise quality judgments on TS blocks. These judgments are then converted into scalar scores using the Bradley-Terry model and used to train an efficient scoring model TSRater composed of a TSFM encoder and an MLP head. Crucially, TSRater is trained using a meta learning approach across multiple domains, enabling few shot adaptation to new, unseen TS data types. The paper demonstrates TSRating's effectiveness through extensive experiments on downstream tasks, showing improved performance when selecting data based on its scores compared to baseline methods.

**Strengths:**

1 The paper exhibits strong originality through its synthesis of multiple concepts: applying the LLM-as-a-judge paradigm cross-modally to raw time series signals, formulating a unique knowledge distillation pipeline to train an efficient scoring model from LLM preferences; and being the first to apply meta-learning for learning a cross-domain data quality scoring function. This architectural combination is novel.

2 The TSRating framework is well-designed. The use of pairwise comparisons, Bradley-Terry modeling, a TSFM encoder, and MAML with SignSGD represents a thoughtful integration of techniques. Experimental validation is extensive across 11 datasets and 3 tasks, providing strong support for the claims.

3 The paper is clearly written and organized. Problem, methodology, and results are presented logically. Prompt design rationale and LLM validation enhance clarity.

4 This work addresses an important gap in time series analysis – scalable and adaptable data quality assessment. As datasets grow in size and diversity, such a tool is valuable. The framework offers a practical approach to automated data curation, potentially improving TSFM training robustness and efficiency.

**Weaknesses:**

1 The framework relies on LLM judgments, which may have inherent biases or sensitivity to prompts. While stability measures were taken , potential systematic biases require further discussion.

2 Justification for selecting the four specific criteria:" trend, frequency, amplitude, pattern" could be stronger. Their sufficiency for all TS tasks (e.g., anomaly detection) is unclear, and potential subjectivity exists.

3 Collecting numerous LLM judgments across multiple domains for meta-learning might be costly and time-consuming, potentially limiting practical scalability.

4 Performance relies on MOMENT's representations. Sensitivity to this choice and effectiveness with other encoders should be discussed.

5 Simple averaging is used for channels and criteria. Exploring adaptive aggregation might improve performance.

**Questions:**

1 Can you elaborate on potential biases in LLM judgments (e.g., pattern preference, positional bias) and mitigation strategies? Did you consider using ensemble judgments from different LLMs?

2 What is the theoretical or empirical basis for selecting these four specific criteria? Are they sufficient for all types of time series tasks (e.g., anomaly detection)? Do you plan to explore more dynamic or task-adaptive criteria?

3 What are the costs and time involved in collecting the LLM judgments for meta-learning? How scalable do you see this approach in practice, especially for covering many more domains or larger datasets?

4 How much does TSRater's performance rely on the MOMENT encoder? What performance changes might be expected if using other TSFM encoders (e.g., Chronos, TimeGPT)?

5 Did you experiment with aggregation methods beyond simple averaging (e.g., weighted average, attention) for fusing scores across channels or criteria?

---

> ### Author Response · Authors · 2025-11-23
>
> Thank you very much for your detailed and constructive comments and suggestions. We provide feedback on each question and concern in a point-to-point manner below.
>
>
> > ##### Re W1 & Q1: The framework depends on LLM judgments that may carry inherent biases or prompt sensitivity. Further discussion on possible systematic biases (e.g., pattern preference, positional bias) and mitigation strategies such as using ensemble judgments from multiple LLMs is needed.
>
> We sincerely appreciate the reviewer’s insightful comments. We fully agree that potential systematic biases in LLM-based judgments are realistic concerns, as large models may exhibit a preference toward certain sequence patterns or positional dependencies in the input. To further elaborate on the robustness of TSRating under these factors, we conduct additional experiments and analysis.
>
> **Potential Biased-Pattern Preference:** We have evaluated several LLMs on a controlled synthetic dataset with known ground-truth characteristics to evaluate whether LLMs inherently prefer certain patterns. The results show that LLMs do exhibit different recognition accuracy across pattern types. At the same time, their overall performance across the four criteria remains consistently high (mostly higher than 0.95). We think this may stem from the fact that these criteria represent fundamental and widely studied properties in time series analysis, making them relatively easier for LLMs to identify.
>
> | Model       | Trend  | Frequency | Amplitude | Pattern |
> | ----------- | ------ | --------- | --------- | ------- |
> | GPT-4o-mini | 0.9525 | 0.9625    | **0.9850**    | 0.9725  |
> | Gemini-2.0  | 0.9575 | 0.9250    | 0.9925    | **0.9975**  |
> | Claude-3.5  | **0.9950** | 0.9425    | 0.9650    | 0.9775  |
> | DeepSeek-V3 | 0.9575 | **1.0000**    | 0.9750    | 0.9700  |
>
> In our TSRating framework, each criterion is scored independently, aiming at mitigating potential systematic preference toward a particular structural pattern.
>
> **Potential Biased-Positional Bias:** Some LLMs might partially depend on the input position of cues when interpreting a sequence. To investigate this, we compared two settings: using a single fixed input order and averaging judgments obtained from two swapped input orders. The results are shown below:
>
> | Criterion | Both Order | Single Order |
> | --------- | ---------- | ------------ |
> | Trend     | 0.9525     | 0.9500       |
> | Frequency | 0.9625     | 0.9550       |
> | Amplitude | 0.9850     | 0.9850       |
> | Pattern   | 0.9725     | 0.9450       |
>
> We observe that averaging across both orders gives slightly better and more stable results, particularly for the Pattern criterion, indicating that positional influence does exist. To mitigate this effect in practice, TSRating has incorporated positional swapping during evaluation so that judgments are averaged across both orders, reducing positional dependency without requiring changes to the downstream scoring process. More details can be found in Section 3.2.
>
> **Ensembled LLM Judgements Evaluation:** Finally, we also explore ensemble judgments across multiple LLMs. In this new experiment on the Electricity dataset, the ensemble strategy first aggregates quality scores from three LLMs and then selects training samples based on the combined judgment. Forecasting models are then trained on these selected samples and compared with the results obtained using individual LLMs. The ensemble approach achieves performance that is comparable to, and in some cases slightly better than, using a single model alone, suggesting that model diversity can contribute to additional stability and robustness in the scoring process.
>
> | Model       | Linear | CNN   | PatchTST | iTransformer | TimeMixer |
> | ----------- | ------ | ----- | -------- | ------------ | --------- |
> | GPT-4o-mini | 1.390  | 1.511 | 0.397    | 0.300        | 0.345     |
> | Gemini-2.0  | 1.410  | 1.501 | 0.402    | 0.304        | 0.323     |
> | Claude-3.5  | 1.406  | 1.508 | 0.404    | 0.307        | 0.334     |
> | Ensemble    | 1.391  | 1.505 | 0.389    | 0.303        | 0.333     |
>
> As a summary, we acknowledge that LLM-based judgments inherently contain biases, such as pattern preference and positional dependencies. Through additional experiments, we find that our design of positional swapping during evaluation and independent scoring can mitigate these biases to some extent. Besides, although ensemble judgments from multiple LLMs can slightly improve the performance in some cases, this does not diminish the significance of the current design, which presents the first attempt to leverage LLM for scalable and human annotation-free time series data rating. We would like to thank the reviewer again for raising these important concerns, which inspire future directions towards more robust and bias-free time series data quality rating with LLMs.

---

> > ### Author Response · Authors · 2025-11-23
> >
> > > ##### Re W2 & Q2: The choice of the four criteria needs stronger justification, as their sufficiency and objectivity across diverse time series tasks (e.g., anomaly detection) remain unclear. Discussion on their theoretical and empirical basis and the potential to explore more dynamic, task-adaptive criteria is needed.
> >
> > We sincerely appreciate the reviewer’s thoughtful questions regarding the selection and sufficiency of the four criteria: trend, frequency, amplitude, and pattern.
> >
> > The choice of these four criteria is supported by both theoretical and empirical considerations. 1) For theoretical consideration, classical time series analysis literature [1,2,3] identifies trend, seasonality (frequency), periodicity, and various patterns as fundamental components that characterize the underlying structure of time series data. These elements are generally regarded in the field as key descriptive dimensions for understanding and modeling temporal dynamics. 2) For empirical consideration, recent studies further support the utilization of these criteria. For example, [4] developed a benchmark to evaluate large language models’ comprehension of time series data using these dimensions as key evaluation axes. Additionally, works such as [5], [6], and [7] have incorporated these characteristics into multi-domain forecasting and generation tasks, demonstrating their practical effectiveness across a range of datasets. More detailed discussions are included in Section 3.3.
> >
> > Regarding their sufficiency for all types of time series tasks, we acknowledge that these four criteria mainly cover mainstream predictive tasks and may not directly capture all specialized needs. At the same time, our TSRating framework is designed to be flexible and extensible, allowing for the incorporation of additional self-defined or task-specific criteria as needed. To elaborate on this point, we provide a new case study experiment by extending TSRating to deal with the anomaly detection task during the rebuttal period. Specifically, we extend the framework by introducing an additional “anomaly” criterion to identify irregular or rare patterns, while still using the original four criteria to select reliable “normal” samples. This case study is conducted on the MSL dataset for binary anomaly detection, and the results measured by F-score across five downstream models demonstrate that TSRating’s combined approach outperforms random selection as well as established data valuation methods like DataShapley and TimeInf:
> >
> > | Method       | Linear     | CNN        | PatchTST   | iTransformer | TimeMixer  |
> > | ------------ | ---------- | ---------- | ---------- | ------------ | ---------- |
> > | Random       | 0.7678     | 0.7953     | 0.7936     | 0.7912       | 0.7897     |
> > | DataShapley  | 0.7780     | 0.8144     | 0.8140     | 0.8125       | 0.8103     |
> > | TimeInf      | 0.7909     | 0.8159     | 0.8140     | 0.8137       | 0.8120     |
> > | **TSRating** | **0.8253** | **0.8480** | **0.8506** | **0.8498**   | **0.8485** |
> >
> > Finally, we acknowledge that exploring more dynamic and task-adaptive criteria is a valuable direction for future work. We envision developing new criteria by summarizing distinctive domain-specific features and integrating them seamlessly into the TSRating framework to better suit diverse downstream applications. We will incorporate the above clarification, the new case study, and the future direction discussion into the revised paper.

---

> ### Author Response · Authors · 2025-11-23
>
> > ##### Re W3 & Q3: Collecting extensive LLM judgments across multiple domains for meta-learning can be costly and time-consuming, which may limit scalability when extending to more domains or larger datasets.
>
> Thank you for your insightful question. For meta-learning, we collected numerous LLM judgments across 9 domains and 22 datasets. The total time spent on generating these judgments was approximately **4,167 seconds**, with an API cost of around **5 USD**. The total token usage for these LLM judgments was about **14.3 million tokens**, with each subset involving roughly equal numbers of pairwise comparisons. The runtime breakdown of TSRater’s components is summarized in the table below. More details are included in Section 4.1.
>
> | Method / Component  | Time (seconds) |
> | ------------------- | -------------- |
> | DataShapley*        | 210,000        |
> | KNNShapley          | 152            |
> | DataOob             | 4,785          |
> | TimeInf             | 4,938          |
> | **TSRater (Total)** | **4,687**      |
> | -- LLM Judgments    | **4,167**      |
> | -- Meta-training    | 323            |
> | -- Few-shot Tuning  | 55             |
> | -- Inference        | 142            |
>
> **Annotation Expense Perspective:** Our LLM-based judgment approach eliminates the need for human annotation of time-series patterns and data quality, which would otherwise incur substantial expenses, especially for certain time series domains required for expert-level annotators. In our current setup, the LLM-based judgments span 9 domains and 22 datasets, covering a considerable amount of time-series data comparable to the corpora used to train existing time-series foundation models. Although LLM-based annotation does involve time and API costs, these remain moderate relative to the expense of human annotation. Moreover, if further scaling is needed, the time and cost associated with LLM-based annotation are likely to remain more feasible than relying on human labeling.
>
> **Generalization Capability Perspective:** We find that TSRater demonstrates promising generalization as well as **few-shot (or zero-shot) scoring** capabilities, enabling it to adapt effectively to new, unseen datasets, which traditional classifier-based models lack. This observation is further supported by the experiment presented in Appendix C.4, Figure 8, which illustrates TSRater’s ability to generalize well to unseen domains. In addition, the time required for few-shot tuning and inference is comparatively small and can be considered as a one-time investment. Therefore, after the initial collection of LLM judgments, expanding TSRater to accommodate new domains typically does not necessitate extensive re-annotation efforts. This design choice contributes to making the approach more scalable and practical.
>
> In summary, although LLM-based judgment introduces LLM inference costs (e.g., API expenses), TSRater’s scalability remains favorable compared to traditional models and human annotations, allowing it to support new domains and datasets with relatively modest additional resources and avoid prohibitive human-annotation costs.

---

> ### Author Response · Authors · 2025-11-23
>
> > ##### Re W4 & Q4: Sensitivity to the use of MOMENT and potential effects when using other TSFM encoders (e.g., Chronos, TimeGPT) should be discussed, including the performance variations.
>
> We thank the reviewer for raising this insightful question. To evaluate the dependence of TSRater on the choice of time series foundation model (TSFM) encoder, we conduct additional ablation experiments comparing three representative TSFMs, including **MOMENT**, **Chronos**, and **TimeGPT**. In this experiment, we apply TSRater with different TSFM encoders for data selection on the Weather dataset and evaluate downstream forecasting performance across five commonly used downstream models, keeping all other experimental settings consistent with those in the main paper.
>
> | Encoder | Linear | CNN   | PatchTST | iTransformer | TimeMixer |
> | ------- | ------ | ----- | -------- | ------------ | --------- |
> | MOMENT  | 0.611  | 0.734 | 0.467    | 0.444        | 0.433     |
> | Chronos | 0.613  | 0.735 | 0.464    | 0.447        | 0.428     |
> | TimeGPT | 0.609  | 0.738 | 0.479    | 0.438        | 0.435     |
>
> The results show that the results with different encoders are consistent and comparable, indicating that TSRater’s effectiveness is not strongly tied to MOMENT or a specific TSFM architecture. Since TSRater operates primarily by learning quality judgments from LLM supervision rather than relying on pretrained representations alone, its performance gain appears relatively independent of the underlying encoder choice.
>
> We also acknowledge that a strong TSFM encoder still contributes positively to the overall performance. When we replaced the MOMENT encoder (109M parameters) with a lightweight Transformer-based T5 encoder (only about 10% of MOMENT’s size), we observed a noticeable degradation in downstream forecasting performance. A summary of the results on the Electricity dataset is shown below (lower is better):
>
> | Method                      | Linear    | CNN       | PatchTST  | iTransformer | TimeMixer |
> | --------------------------- | --------- | --------- | --------- | ------------ | --------- |
> | **TSRating with MOMENT**    | **1.390** | **1.511** | **0.397** | **0.300**    | **0.345** |
> | **TSRating with T5 encoder** | 1.541     | 1.534     | 0.419     | 0.307        | 0.366     |
>
> These results suggest that while TSRater does not rely on the specific identity of MOMENT, having a sufficiently expressive encoder is beneficial for extracting informative representations for quality assessment. More detailed results and analysis can be found in Table 15 in Appendix D.5.

---

> ### Author Response · Authors · 2025-11-23
>
> > ##### Re W5 & Q5: The method uses simple averaging for channels and criteria, while adaptive aggregation (e.g., weighted average, attention) is worth further exploration.
>
>
> We sincerely appreciate the reviewer’s constructive suggestion. Exploring **adaptive aggregation strategies** for both criteria and channels (e.g., weighted averaging, attention-based fusion, or task-adaptive mechanisms) is indeed meaningful and may further improve TSRating’s effectiveness in certain scenarios. We fully agree that this represents a valuable direction, and we have highlighted it as part of our future work. Motivated by this insight, we conduct two additional sets of experiments to more concretely examine how alternative aggregation strategies on criteria and channels could influence quality rating performance.
>
>
> ### (1) Adaptive Aggregation Across Criteria
>
> We conduct a **weighted criterion fusion experiment** on the Weather dataset, where the weights were assigned based on domain understanding of weather time series (Trend = 0.30, Frequency = 0.10, Amplitude = 0.25, Pattern = 0.35). These weights reflect that trend and pattern often capture key structural behavior, amplitude captures the strength of change, and frequency receives a lower weight due to sensitivity to noise and redundancy with other criteria. The downstream forecasting RMSE results (using the top 50% selected samples) are as follows:
>
> |                  | Linear | CNN   | PatchTST | iTransformer | TimeMixer |
> | ---------------- | ------ | ----- | -------- | ------------ | --------- |
> | Simple average   | 0.611  | 0.734 | 0.467    | 0.444        | 0.433     |
> | Weighted average | 0.611  | 0.719 | 0.473    | 0.456        | 0.424     |
>
> As shown, the results across different models are quite close, with neither strategy providing a consistently dominant advantage. This suggests that in this setting, simple averaging is already sufficient for combining criterion scores.
>
> Beyond this experiment, we also plan to explore more adaptive and dynamic mechanisms for weighting the four criteria. Instead of relying on manually defined weights, future work could treat the criterion weights as learnable parameters during meta-rater training. Concretely, the meta-rater can produce scoring predictions based on a weighted aggregation of criterion-specific representations, where the weights are initialized uniformly but optimized through backpropagation by minimizing the validation loss across multiple tasks and domains. In this way, the learning process encourages the system to automatically adjust the importance of each criterion in response to the characteristics of each domain, dataset, or task. This would allow weights to emerge from theoretical optimization rather than manual prior knowledge, and might potentially yield a more robust and domain-agnostic scoring mechanism.

---

> ### Author Response · Authors · 2025-11-23
>
> > ##### [continue] Re W5 & Q5 ....
>
> ### (2) Adaptive Aggregation Across Channels
>
> We also add a new **weighted aggregation across channels** experiment on Weather dataset, with weights assigned based on meteorological prior knowledge:
>
> | Channel     | Weight | Rationale (brief)                                     |
> | ----------- | ------ | ----------------------------------------------------- |
> | Temperature | 0.40   | Major driver of large-scale weather dynamics          |
> | Humidity    | 0.25   | Strong relevance to cloud formation and precipitation |
> | Wind        | 0.20   | Important but more variable in stability              |
> | Radiation   | 0.15   | Sparser and more task-dependent                       |
>
> Forecasting performance is shown below:
>
> |                  | Linear | CNN   | PatchTST | iTransformer | TimeMixer |
> | ---------------- | ------ | ----- | -------- | ------------ | --------- |
> | Simple average   | 0.620  | 0.720 | 0.480    | 0.471        | 0.545     |
> | Weighted average | 0.622  | 0.731 | 0.476    | 0.479        | 0.532     |
>
> The differences across different weighting strategies are modest and do not show consistent advantages, indicating that weighting can shift the scoring outcomes slightly but does not necessarily translate into systematic improvement in this scenario.
>
> In future work, we also plan to explore a more dynamic and learnable strategy for channel weighting, drawing inspiration from Mixture-of-Experts (MoE) architectures. Rather than using manually assigned weights, a small gating network can be trained to produce importance scores for each channel based on the input data and validation feedback during training. These scores are then used to weight each channel’s output, allowing the aggregated representation to dynamically emphasize the channels that most effectively contribute to downstream performance in a data-driven way. This approach allows weights to be automatically adapted across datasets and domains, and may lead to more robust and generalizable channel aggregation in future extensions of TSRating.
>
> In summary, we appreciate the reviewer’s valuable suggestions. While simple averaging already performs well in our current experiments, we agree that more adaptive fusion mechanisms, such as learnable weighting, gating mechanisms, or attention, represent promising future extensions and may provide additional benefits in tasks where channel or criterion importance varies more significantly. We will incorporate the above new experiments and future work discussions into the revised paper.
>
>
> > **Additional References**
>
> [1]Robert B Cleveland, William S Cleveland, Jean E McRae, Irma Terpenning, et al. Stl: A seasonal-trend decomposition. J. off. Stat, 6(1):3–73, 1990.
>
> [2]Rob J Hyndman and George Athanasopoulos. Forecasting: principles and practice. OTexts, 2018.
>
> [3]Seyed Mehran Kazemi, Rishab Goel, Sepehr Eghbali, Janahan Ramanan, Jaspreet Sahota, Sanjay Thakur, Stella Wu, Cathal Smyth, Pascal Poupart, and Marcus Brubaker. Time2vec: Learning a vector representation of time. arXiv preprint arXiv:1907.05321, 2019.
>
> [4]Yifu Cai, Arjun Choudhry, Mononito Goswami, and Artur Dubrawski. Timeseriesexam: A time series understanding exam. arXiv preprint arXiv:2410.14752, 2024.
>
> [5]Yu-Hao Huang, Chang Xu, Yueying Wu, Wu-Jun Li, and Jiang Bian. Timedp: Learning to generate multi-domain time series with domain prompts. In Proceedings of the AAAI Conference on Artificial Intelligence, volume 39, pp. 17520–17527, 2025.
>
> [6]Mononito Goswami, Konrad Szafer, Arjun Choudhry, Yifu Cai, Shuo Li, and Artur Dubrawski. Moment: A family of open time-series foundation models. arXiv preprint arXiv:2402.03885, 2024.
>
> [7]Gerald Woo, Chenghao Liu, Akshat Kumar, Caiming Xiong, Silvio Savarese, and Doyen Sahoo. Unified training of universal time series forecasting transformers. arXiv preprint arXiv:2402.02592, 2024.

---

> ### Comment · Reviewer_pUfp · 2025-11-27
>
> I thank the authors for their comprehensive rebuttal, my main concerns have been addressed. Based on the additional materials and experiments provided in the response, I consider this a solid research work. I am happy to raise my score to 8.

---

> > ### Author Response · Authors · 2025-11-27
> >
> > Thank you very much for your thorough review and encouraging feedback. We sincerely appreciate your constructive comments that help us strengthen our work. We are truly grateful for your positive assessment and are delighted that our response has addressed your concerns.

---

### Official Review · Reviewer_kv5d · 2025-10-31

**Soundness:** 3
**Presentation:** 3
**Contribution:** 4
**Rating:** 4
**Confidence:** 3

**Summary:**

This paper introduces TSRating, a framework designed to evaluate the quality of time series data using meta-learning guided by large language models (LLMs). TSRating identifies quality based on four criteria: trend, frequency, amplitude, and pattern. It leverages LLMs to compare pairs of time series and convert these judgments into quality predictions via the TSRater model, which adapts across multiple domains. The framework employs meta learning for efficient training and demonstrates superior performance over traditional methods in experiments. TSRating is shown to be effective in identifying high-quality time series samples, enhancing model performance in various forecasting and classification tasks.

**Strengths:**

1. Proposes an innovative method for time series quality evaluation by employing LLMs as general “pattern evaluators”, moving beyond traditional statistical or contribution-based metrics.
2. Uses meta-learning to effectively address domain heterogeneity, resulting in strong cross-domain generalization and applicability.
3. Clearly defines the evaluation criteria and provides transparent methodology.
4. Demonstrates significant practical relevance through measurable improvements in downstream tasks.

**Weaknesses:**

1. Heavy reliance on LLM judgments may introduce biases or inaccuracies in understanding intrinsic time series patterns, risking propagation of these biases into TSRater.
2. The definition of “quality” along four fixed dimensions implicitly equates high quality with signal clarity or prominence, which may not be optimal for all downstream tasks (e.g., classification vs. forecasting), potentially leading to suboptimal data selection.

**Questions:**

1. Given that real-world time series data may contain rare or complex patterns that score poorly yet are crucial for future events, how does TSRating avoid discarding “anomalous but important” sequences?
2. Could TSRater exhibit a preference for common or simple patterns, thus reducing diversity in the selected training set and harming generalization? For example, in Figure 4(a) indices 1531 and 1552, and in Figure 4(c) indices 860 and 883, there are long overlapping segments.
3. How does the choice of data segmentation strategy (e.g., different sequence lengths or block sizes) affect TSRater’s evaluation performance?
4. Regarding the integration of multiple criteria (trend, frequency, amplitude, pattern), what is the specific fusion method used? Can a fixed fusion strategy generalize well to all types of downstream tasks?

If the authors are able to address the questions raised, I would be inclined to increase my evaluation score.

---

> ### Author Response · Authors · 2025-11-23
>
> We sincerely thank the reviewer for your valuable comments and insightful suggestions. Below is our point-by-point response to the concerns raised.
>
> > ##### Re W1: Heavy Reliance on LLM judgments may introduce biases and inaccuracies into TSRater
>
> We sincerely thank the reviewer for raising the important point regarding potential biases and inaccuracies in LLM judgments when understanding intrinsic time series patterns. We fully acknowledge this concern and conduct additional experiments to explore and mitigate these issues, while also providing evidence of the LLM’s capability in this domain.
>
> To begin with, we compared the performance of LLM judgments against traditional time series pattern recognition methods [1]. On synthetic datasets, we train a traditional supervised Support Vector Machine (SVM) classifier to recognize the four quality criteria—Trend, Frequency, Amplitude, and Pattern. The results suggest that LLM models such as GPT-4o-mini, Gemini-2.0, and Claude-3.5, although present unavoidable inaccuracies, tend to perform better than the traditional approach in identifying these key time series features, indicating the potential of LLM-based judgments.
>
> | Criterion  | GPT-4o-mini | Gemini-2.0 | Claude-3.5 | SVM    |
> |------------|-------------|------------|------------|--------|
> | Trend      | 0.9525      | 0.9575     | 0.9950     | 0.9275 |
> | Frequency  | 0.9625      | 0.9250     | 0.9425     | 0.9350 |
> | Amplitude  | 0.9850      | 0.9925     | 0.9650     | 0.9550 |
> | Pattern    | 0.9725      | 0.9975     | 0.9775     | 0.9125 |
>
> This promising performance of LLM-based judgements compared to the traditional SVM approach could be attributed to our design of bias mitigation. Below, we outline three key components for mitigating bias in our TSRating design.
>
> Firstly, instead of relying on direct numerical scores that might introduce biases such as subjective interpretation of the scale, we employ pairwise comparisons. This approach asks the model to judge which of two samples is of higher quality according to the judgment criteria rather than assigning an absolute score. Experimental results using GPT-4o-mini demonstrate that using pairwise comparisons improves the accuracy of pattern recognition to some extent. More details can be found in Appendix D.5.
>
> | Criterion  | LLM (Pairwise) | LLM (Direct Score) |
> |------------|----------------|--------------------|
> | Trend      | 0.995          | 0.831              |
> | Frequency  | 1.000          | 0.862              |
> | Amplitude  | 0.998          | 0.871              |
> | Pattern    | 0.968          | 0.834              |
>
> Secondly, we incorporate multiple voting mechanisms to reduce variance from individual LLM judgments. Our experiments on synthetic datasets show that aggregating multiple votes consistently yields better accuracy compared to single votes. In our experiments, we set the number of LLM comparisons to 20. Further details are described in Section 3.2.
>
> | Criterion  | Multiple Vote | Single Vote |
> |------------|---------------|-------------|
> | Trend      | 0.9525        | 0.9050      |
> | Frequency  | 0.9625        | 0.9200      |
> | Amplitude  | 0.9850        | 0.9750      |
> | Pattern    | 0.9725        | 0.8975      |
>
> Thirdly, to mitigate order bias in pairwise comparisons, we compare results when pairs are judged in both orders versus a single order. Experimental results show that allowing judgments in both directions improves accuracy on several criteria. Further details are also described in Section 3.2.
>
> | Criterion  | Both Order | Single Order |
> |------------|------------|--------------|
> | Trend      | 0.9525     | 0.9500       |
> | Frequency  | 0.9625     | 0.9550       |
> | Amplitude  | 0.9850     | 0.9850       |
> | Pattern    | 0.9725     | 0.9450       |
>
> Besides, we have conducted two sets of validation experiments, one on real-world datasets and the other on synthetic datasets, aiming to demonstrate that LLMs could capture intrinsic time series patterns relevant for quality assessment. These results provide empirical support for the effectiveness of LLM-based judgments in both practical and controlled environments. More details are available in Appendix B.
>
> In summary, while we recognize that systematic biases and inaccuracies in LLM judgments may exist, our experiments suggest that carefully designed mitigation strategies can meaningfully reduce their impact. Compared to traditional manual annotation approaches, using LLMs provides a scalable, less subjective, and more effective means of assessing time series data quality. We appreciate the reviewer’s valuable comments, which helped us clarify and further validate this aspect of our work.

---

> > ### Author Response · Authors · 2025-11-23
> >
> > > ##### Re W2: Fixed four-dimension definition of “quality” may not be optimal for all downstream tasks
> >
> > We thank the reviewer for raising this important point. We acknowledge that the fixed four criteria of “quality” (trend, frequency, amplitude, and pattern) may not be optimal for all downstream tasks. Indeed, different tasks may require different data characteristics to achieve the best performance. At the same time, our TSRating framework is designed to be flexible and extensible (with competitive performance on common time series tasks like classification and forecasting), allowing easy integration of additional quality criteria beyond the initial four criteria. For example, we experiment with incorporating an additional criterion — **stationarity**, which measures the stability of time series over time. Experimental results on four forecasting datasets using TimeMixer (as a downstream model) show that adding this criterion improves performance on Electricity and Traffic, but not consistently across all testing datasets:
> >
> > | Dataset     | 4 Criteria | 4 Criteria + Stationary |
> > | ----------- | ---------- | ---------------------- |
> > | Weather     | 0.433      | 0.444                  |
> > | Electricity | 0.318      | **0.312**                  |
> > | Exchange    | 0.223      | 0.227                  |
> > | Traffic     | 0.285      | **0.276**                  |
> >
> > Furthermore, we conduct ablation studies by removing each of the original four criteria individually to examine their necessity. The results generally show performance degradation after removing any single criterion, suggesting that all four contribute meaningfully to the overall quality assessment under the current framework. More details about these experiments can be found in Section 4.2.
> >
> > | Dataset     | 4 Criteria | -Trend | -Frequency | -Amplitude | -Pattern |
> > | ----------- | ---------- | ------ | ---------- | ---------- | -------- |
> > | Weather     | **0.433**      | 0.440  | 0.436      | 0.438      | 0.442    |
> > | Electricity | 0.318      | 0.333  | 0.326      | **0.315**      | 0.319    |
> > | Exchange    | **0.223**      | 0.228  | 0.230      | 0.234      | 0.242    |
> > | Traffic     | **0.285**      | **0.285**  | 0.295      | 0.293      | 0.289    |
> >
> > These results suggest that while our initial choice of criteria captures essential aspects of time series quality, the framework itself is flexible enough to incorporate additional user-defined or task-specific criteria as needed, and these extensions can be tailored depending on the downstream application to achieve the best selection performance. We will also clarify this flexibility in the revised manuscript.

---

> > > ### Author Response · Authors · 2025-11-23
> > >
> > > > ##### Re Q1: How does TSRating avoid discarding anomalous but important anomalous sequences?
> > >
> > > We appreciate the reviewer for highlighting the important issue of rare or complex patterns in real-world time series data, which may be scored poorly yet hold critical significance for downstream tasks. We fully acknowledge that our current four quality criteria in TSRating are designed for a general-purpose predictive setting and do not explicitly target such rare or anomalous patterns. Specifically, the four criteria are designed and evaluated to capture regularity patterns that are known to be crucial for long-/short-term forecasting and classification tasks, which are fundamental and pervasive time-series applications (and are the primary focus of nearly all current TSFMs). That being said, we also demonstrate below that the TSRating framework is flexible enough to incorporate additional criteria beyond the current four criteria, enabling TSRating to explicitly evaluate anomalous yet important patterns required for the anomaly detection task.
> > >
> > > **A New Case Study on Anomaly Detection:** We conduct a new case study focusing on anomaly detection, where “anomalous but important” sequences are crucial for such a task. Specifically, we augment our framework by designing an additional task-specific criterion-“anomaly” aimed at identifying irregular or anomalous patterns, while continuing to use the original four criteria to recognize reliable “normal” patterns. We evaluate this extended TSRating framework on the MSL dataset for a binary anomaly detection task.
> > >
> > > Our experimental results, reported in terms of F-score across five downstream models (Linear, CNN, PatchTST, iTransformer and TimeMixer), demonstrate that the samples selected by TSRating’s extended rating mechanism outperform random selection as well as existing data valuation methods such as DataShapley and TimeInf:
> > >
> > > | Method       | Linear     | CNN        | PatchTST   | iTransformer | TimeMixer  |
> > > | ------------ | ---------- | ---------- | ---------- | ------------ | ---------- |
> > > | Random       | 0.7678     | 0.7953     | 0.7936     | 0.7912       | 0.7897     |
> > > | DataShapley  | 0.7780     | 0.8144     | 0.8140     | 0.8125       | 0.8103     |
> > > | TimeInf      | 0.7909     | 0.8159     | 0.8140     | 0.8137       | 0.8120     |
> > > | **TSRating** | **0.8253** | **0.8480** | **0.8506** | **0.8498**   | **0.8485** |
> > >
> > > These results indicate that TSRating is flexible enough to incorporate task-specific criteria tailored to emphasize irregularities, where it effectively preserves important anomalous sequences while maintaining robust selection of high-quality data. Visualization examples supporting this analysis are shown in Appendix B (Figure 7). We believe this flexibility of TSRating to adapt new criteria is a key strength in handling specific downstream applications.

---

> > > > ### Author Response · Authors · 2025-11-23
> > > >
> > > > > ##### Re Q2: Could TSRater favor common/simple patterns (seen in Figure 4) and reduce data diversity?
> > > >
> > > > We sincerely thank the reviewer for the careful and insightful observation regarding the overlapping segments shown in Figure 4. These overlapping segments are shown as examples to illustrate the block-level scoring mechanism within our framework, where each time series is divided into overlapping blocks that are pairwise compared by the LLM based on quality criteria, as described in Section 3.2 and formalized by Equation (1). Since TSRating segments the time series data using overlapping windows, blocks with high scores are also presented as overlapping segments, reflecting the inherent continuity of the underlying patterns. We would like to emphasize that this overlap at the block level does not necessarily indicate a reduction in diversity at the sample level across the whole dataset. To provide further details and avoid confusion, we have included additional visual analyses in Figure 5 and Figure 6 om Appendix B.2.
> > > >
> > > > To further address the concern about potential reduction in diversity, we conduct a new experiment for comparative analysis of diversity between the top 50% high-quality data selected by TSRating and random selection across four datasets. We used the distance-based diversity metric following [2], which quantifies the diversity of a dataset by aggregating pairwise distances between samples, where larger values indicate higher diversity. The results are summarized as follows:
> > > >
> > > > | Dataset    | Weather | Electricity | Exchange | Traffic |
> > > > |------------|---------|-------------|----------|---------|
> > > > | Random     | 0.812   | 0.698       | 0.675    | 0.733   |
> > > > | TSRating   | 0.805   | 0.685       | 0.670    | 0.725   |
> > > >
> > > > Our experiments suggest that the data selected by TSRating maintains diversity levels comparable to those of random selection, which provides evidence of the framework’s ability to preserve the data diversity and thus indicates robustness and potential for generalization. Additionally, we also observe improved generalization performance on these datasets, as reported in Table 1 of the manuscript. We would like to thank the reviewer again for raising the concern about data diversity.
> > > >
> > > > > ##### Re Q3: How sensitive is TSRater to the choice of segmentation strategy (e.g., block size or sequence length)?
> > > >
> > > > We sincerely thank the reviewer for raising the important question regarding the impact of data segmentation strategies on TSRating’s evaluation performance. In our design, the segmentation of time series data into fixed-length blocks aims to balance capturing sufficient temporal context while maintaining manageable complexity for pattern recognition. Different block sizes can influence the granularity of features captured and thus potentially affect the quality assessment and downstream forecasting performance.
> > > >
> > > > To investigate this, we have conducted experiments on the Traffic dataset using the TimeMixer as the downstream model, comparing TSRating with several baseline selection methods under three different block sizes: 128(our choice), 288, and 432. More details can be found in Appendix D.6.
> > > >
> > > > | Method       | 128      | 288       | 432       |
> > > > | ------------ | --------- | --------- | --------- |
> > > > | Random       | 0.311     | 0.308     | 0.313     |
> > > > | DataOob      | 0.317     | 0.310     | 0.311     |
> > > > | DataShapley  | 0.302     | 0.295     | 0.297     |
> > > > | KNNShapley   | 0.285     | 0.393     | 0.308     |
> > > > | TimeInf      | 0.308     | 0.296     | 0.291     |
> > > > | **TSRating** | **0.282** | **0.285** | **0.289** |
> > > >
> > > >
> > > > The results indicate that TSRating maintains competitive forecasting performance across different block sizes. Larger block sizes (432) do not significantly improve performance, suggesting that an excessively large temporal window may not bring additional benefits for quality evaluation in this setting. Therefore, maintaining a moderate block size appears to be more effective by balancing the need to capture meaningful temporal patterns while avoiding unnecessary complexity. In summary, these findings confirm that TSRating is robust to changes in block size and can handle different segmentation strategies without significant loss of performance.

---

> > > > > ### Author Response · Authors · 2025-11-23
> > > > >
> > > > > > ##### Re Q4: How are the four criteria fused, and can a fixed fusion strategy generalize across tasks?
> > > > >
> > > > > We appreciate the reviewer’s insightful question regarding the fusion method for integrating the four criteria in TSRating. Currently, we adopt a simple uniform average approach to combine these criteria. Based on our experiments, this task-agnostic, stable, and hyperparameter-free method has demonstrated reasonable robustness and generalizability across the downstream tasks we evaluated.
> > > > >
> > > > > At the same time, we acknowledge that a fixed fusion strategy may not be optimal for every possible scenario or task. It is indeed a worthwhile research direction to explore more adaptive fusion methods that can more effectively capture the task-specific importance of each criterion, just as you suggested.
> > > > > For example, in an additional experiment on the Weather dataset, we apply a **weighted average** fusion based on expert knowledge, assigning weights as follows: Trend = 0.30, Frequency = 0.10, Amplitude = 0.25, and Pattern = 0.35. These weights reflect the core characteristics of weather data, emphasizing Pattern due to its representation of underlying physical processes, while giving lower weight to Frequency given to its noise sensitivity. The RMSE results of downstream forecasting models trained on the top 50% high-quality samples selected by TSRating are reported below:
> > > > >
> > > > > | Method       | Simple Averaging | Weighted Averaging |
> > > > > | ------------ | ---------------- | ------------------ |
> > > > > | Linear       | 0.611            | 0.611              |
> > > > > | CNN          | 0.734            | 0.719              |
> > > > > | PatchTST     | 0.467            | 0.473              |
> > > > > | iTransformer | 0.444            | 0.456              |
> > > > > | TimeMixer    | 0.433            | 0.424              |
> > > > >
> > > > > The results show that while weighted averaging may bring marginal improvements in certain models, the overall performance differences remain minor. This suggests that our current simple averaging approach is already effective and generalizes well across downstream tasks.
> > > > >
> > > > > We agree that it is promising to investigate more **adaptive fusion strategies**, where the four criterion scores could be treated as variables optimized dynamically per task. Specifically, we plan to explore approaches such as meta-learning or attention mechanisms to learn task-dependent weights, enabling the fusion process to better align with specific downstream objectives. Although such extensions could further enhance performance, they do not diminish the novelty or significance of the current TSRating framework and its demonstrated capabilities. We hope this clarifies the rationale behind our current fusion method and our openness to future improvements.
> > > > >
> > > > >
> > > > > > **Additional References**
> > > > >
> > > > > [1]Lin J, Williamson S, Borne K, et al. Pattern recognition in time series[J]. Advances in machine learning and data mining for astronomy, 2012, 1(617-645): 3.
> > > > >
> > > > > [2]Drosou M, Jagadish H V, Pitoura E, et al. Diversity in big data: A review[J]. Big data, 2017, 5(2): 73-84.

---

> > > > > > ### Comment · Reviewer_kv5d · 2025-11-26
> > > > > >
> > > > > > I appreciate the authors' efforts in addressing my concerns during the rebuttal phase. Based on the clarifications and additional results provided, I have decided to raise my score to 6.
> > > > > >
> > > > > > Additionally, regarding Figure 3, the current layout results in significant whitespace. It would be beneficial to reconsider the arrangement of the subplots to achieve a more balanced visual effect.

---

> > > > > > > ### Author Response · Authors · 2025-11-26
> > > > > > >
> > > > > > > Thank you again for your careful review and constructive comments. We sincerely appreciate your feedback, which has greatly helped us improve our paper, and we are glad that our response has addressed your concerns. Following your suggestion regarding the visual balance in Figure 3, we have updated the figure in the revised PDF by adding a new subplot with experiments on the MOMENT model and reorganizing the layout to reduce unnecessary whitespace.

---

### Official Review · Reviewer_nhuS · 2025-10-31

**Soundness:** 2
**Presentation:** 3
**Contribution:** 2
**Rating:** 4
**Confidence:** 5

**Summary:**

The paper proposes TSRating, which uses LLM pairwise judgments on four TS criteria including trend, frequency, amplitude and pattern, to score fixed-length blocks via a Bradley–Terry model, then trains a meta-learned TSRater to predict quality efficiently on new datasets. Meta-training relies on MAML with signSGD over 9 domains; in downstream selection, the top-50 percent “high-quality” samples train forecasting/classification models. The authors report improved performance and favorable amortized runtime vs. Shapley/Influence baselines.

**Strengths:**

**S1.** This paper is well-organized and easy to follow.

**S2.** The task of time series datasets quality evaluation is important in deep time series model training.

**S3.** Meta-learning across 9 domains/22 subsets and reuse on unseen datasets is a sensible design for diverse time series.

**Weaknesses:**

**W1.** The most critical weakness lies in the evaluation strategy of TSRating. The framework implicitly assumes that time series with strong trends, seasonality, and regular patterns (Figure 4 in the Appendix) represent high-quality data because they are more predictable and easier to learn. Conversely, series with irregular or unpredictable fluctuations (e.g., the red block in the right part of Figure 1) are treated as “bad samples.” However, in real-world applications, data often exhibit irregular noise and unpredictable variations rather than such idealized patterns. Therefore, the evaluation strategy based on these four criteria is limited in its applicability to real-world scenarios.

**W2.** The LLM judgement is limited, inlcluding:

* Based on LLM pairwise labels yet provides only small-scale synthetic checks and a single-dataset tri-LLM comparison.

* There is no inter-LLM agreement, prompt-order/wording sensitivity, or human/expert adjudication beyond illustrative plots.

* The Bradley–Terry likelihood carries no uncertainty calibration and no analysis of label noise propagation into TSRater.

**W3.** There is no ablation replacing MOMENT with other (TSFM) encoders or fine-tuning MOMENT, so the gains might stem from encoder pretraining/domain coverage rather than the rating scheme. This also risks domain-shift leakage if MOMENT has seen similar sources during pretraining

**W4.** Selecting the top-50 percent across all methods may advantage rank-based raters..

**W5.** Runtime Table 2 aggregates “LLM judgments” but does not report token counts, pairwise comparison budget per dataset, or prompting policy vs. quality.

**W6.** Line 246, synthetic set with obvious hand-crafted time series characteristics labeled as high-quality blocks is not make sense (refer to W1).

**Questions:**

Please see Weaknesses.

Additional questions:

**Q1.** How consistent are the LLM pairwise judgments across prompts, random seeds, and different LLMs?

**Q2.** Does TSRater still perform well if the MOMENT encoder is fine-tuned or replaced with another TSFM backbone?

---

> ### Author Response · Authors · 2025-11-23
>
> We sincerely appreciate your thoughtful comments and constructive suggestions. Below, we provide a detailed response addressing each point raised.
>
> > ##### Re W1: The main limitation is TSRating’s evaluation strategy, which assumes series with strong trends, seasonality, and regular patterns are high-quality, while irregular or unpredictable data are labeled “bad samples.” This may limit applicability in real-world scenarios where noise and irregularities are common.
>
> We sincerely thank the reviewer for the thoughtful comment and fully agree that real-world time series often exhibit irregular noise and unpredictable variations, and that such less regular samples can be important in some real-world scenarios. In the following, we clarify the applicability of TSRating from three perspectives: 1) Additional experiments on a new real-world dataset, 2) Broader use cases of TSRating’s sample ordering of the dataset to implicitly identify irregular samples, 3) Extending TSRating to explicitly capture irregular patterns and a new case study, and 4) Revision and clarification of Figures 1 and 4.
>
> First, we have conducted further evaluations on a real-world dataset called Weather. This dataset is characterized by its high level of noise and irregular variations due to the inherent complexity of weather patterns, sensor measurement errors, and occasional missing data. Statistical analysis reveals that the Weather dataset contains more stochastic fluctuations than other benchmark datasets used in our experiments, including Electricity, Traffic, and ExchangeRate. Detailed comparisons and results using the noise estimation method [1] are presented below.
>
> | Dataset      | Noise Level  |
> |--------------|--------------|
> | Weather      | 0.32         |
> | Electricity  | 0.20         |
> | Traffic      | 0.17         |
> | ExchangeRate | 0.28         |
>
>
> To verify the applicability of TSRating in such realistic and challenging scenarios, we selected the top-rated samples from this dataset to train five different downstream forecasting models. We then compared TSRating’s performance with multiple alternative sample-selection strategies. The results are shown below:
>
> | Method      | Linear | CNN   | PatchTST | iTransformer | TimeMixer |
> |-------------|--------|-------|----------|--------------|-----------|
> | Random      | 0.665  | 0.769 | 0.474    | 0.509        | 0.537     |
> | DataOob     | 0.638  | 0.737 | 0.558    | 0.569        | 0.496     |
> | DataShapley | 0.638  | 0.767 | **0.457** | 0.452      | 0.466     |
> | KNNShapley  | 0.625  | 0.763 | 0.553    | 0.489        | 0.517     |
> | TimeInf     | 0.616  | 0.758 | 0.510    | 0.454        | 0.461     |
> | **TSRating** | **0.611** | **0.734** | 0.467  | **0.444** | **0.433** |
>
> The above results show that even on datasets with real-world noise and irregular variation, TSRating remains competitive and provides stable improvements in downstream forecasting performance. Furthermore, as demonstrated in our evaluation across three task types on eleven benchmark datasets (detailed in Table 10), all of which are derived from real-world data, higher-rated samples consistently lead to better performance for tasks and models that depend on relatively more regular time-series structures, such as long-/short-term forecasting and classification. This indicates that top-rated samples are indeed beneficial for these particular types of tasks and models. A plausible explanation, as suggested by the reviewer, is that these samples exhibit clearer patterns, such as trends and seasonality, which facilitate learning for such tasks. In summary, in application scenarios where relatively high regularity is desired, TSRating can more accurately identify such samples and thereby better support the learning of downstream tasks and models.
>
> Second, we would like to clarify that the ordering of the dataset induced by TSRating's rating scores can be used for purposes beyond selecting only the top-rated samples. As you suggest, when the goal is to identify irregular or noisy samples with low conformity to the four criteria, TSRating can also be used to locate such samples at the lower end of the scoring spectrum. In this sense, TSRating offers a meaningful ordering of rated time-series samples according to their relative regularity under the four criteria, **implicitly** enabling the selection and utilization of samples for tasks and models that require irregular noise or unpredictable variations, in addition to the forecasting and classification tasks discussed above. We will incorporate this discussion into the revised paper.

---

> > ### Author Response · Authors · 2025-11-23
> >
> > > #### [Continue] Re: W1 ....
> >
> > Third, the main contribution of our paper lies in exploring the promising direction of LLM-based time-series rating and in developing an end-to-end pipeline that can accommodate any desired criteria beyond the four criteria reported in the paper. Based on this new and flexible framework, we present a new case study demonstrating that TSRating is flexible enough to incorporate additional criteria when irregular patterns need to be **explicitly** rated. Specifically, this new case study focuses on the anomaly detection task, where recognizing “anomalous but important” sequences is crucial. We augmented our framework by designing an additional criterion “anomaly” aimed at identifying irregular or anomalous patterns, while continuing to use the original four criteria to select reliable “normal” samples. We evaluate on the MSL dataset for a binary anomaly detection task. Our experimental results, reported in terms of F-score across five downstream models (Linear, CNN, PatchTST, iTransformer, and TimeMixer), demonstrate that TSRating’s extended rating outperforms random selection as well as existing data valuation methods such as DataShapley and TimeInf:
> >
> > | Method       | Linear     | CNN        | PatchTST   | iTransformer | TimeMixer  |
> > | ------------ | ---------- | ---------- | ---------- | ------------ | ---------- |
> > | Random       | 0.7678     | 0.7953     | 0.7936     | 0.7912       | 0.7897     |
> > | DataShapley  | 0.7780     | 0.8144     | 0.8140     | 0.8125       | 0.8103     |
> > | TimeInf      | 0.7909     | 0.8159     | 0.8140     | 0.8137       | 0.8120     |
> > | **TSRating** | **0.8253** | **0.8480** | **0.8506** | **0.8498**   | **0.8485** |
> >
> >
> > The above results suggest that by incorporating task-specific criteria tailored to detect irregularities, TSRating can effectively preserve important anomalous sequences while maintaining robust selection of high-quality data. We believe this flexibility of TSRating to adapt new criteria is a key strength in handling specific downstream applications.
> >
> > **Regarding Figure 1:** We also acknowledge that the examples in Figure 1 may unintentionally give the impression that TSRating are only applicable to overly smooth or perfectly structured data. In fact, the choice of examples in Figure 1 was intended only for visual contrast and to make human interpretation easier. We have updated Figure 1 in the revision with time-series samples from a real-world benchmark dataset to avoid causing such confusion.
> >
> > **Regarding Figure 4 and visualization results:** As the reviewer observes, the visualization examples provided in Figure 4 of the original Appendix only present relatively ideal cases with clear repetitive patterns. These samples were originally selected for visualization analysis mainly to make it easier for humans to observe the patterns, but we now realize that such samples do not fully capture the variety present in real datasets. Therefore, we have further expanded Appendix B.2 by adding 20 additional samples for visualization analysis. These new samples contain more realistic noise and irregularities while still exhibiting meaningful structures. According to the visualization analysis, the samples identified as higher-quality under TSRating still tend to display relatively stronger trends, seasonality, and structural regularity compared with lower-quality samples.

---

> > > ### Author Response · Authors · 2025-11-23
> > >
> > > > ##### Re W2 & Q1: LLM judgments have limitations including limited validation, no inter-LLM agreement or human adjudication, and lack of uncertainty analysis; how consistent are pairwise judgments across prompts, seeds, and different LLMs?
> > >
> > > Thank you for your valuable suggestions. Following your suggestion, we conduct additional experiments and analyses to more comprehensively evaluate the consistency, robustness, and reliability of LLM pairwise judgments across different prompts, parameter configurations, and LLM models.
> > >
> > > **Larger synthetic dataset evaluation:** We expand beyond the initial single-dataset tri-LLM comparison and run controlled experiments on a larger-scale synthetic dataset designed to isolate the four quality criteria (trend, frequency, amplitude, pattern). The synthetic dataset is generated by systematically varying templates, combining base patterns (e.g., sine waves, linear trends, AR processes) with noise and anomaly injection in a scalable pipeline inspired by prior work [2]. This dataset includes 4,000 pairwise comparisons with clear ground truth differences. Results from four representative LLMs (GPT-4o-mini, Gemini-2.0, Claude-3.5, DeepSeek-V3) suggest that LLMs generally identify these interpretable structures reliably in idealized conditions:
> > >
> > > | Quality Criterion | GPT-4o-mini | Gemini-2.0 | Claude-3.5 | DeepSeek-V3 |
> > > | ----------------- | ----------- | ---------- | ---------- | ----------- |
> > > | Trend             | 0.953       | 0.958      | 0.989      | 0.956       |
> > > | Frequency         | 0.967       | 0.924      | 0.942      | 0.997       |
> > > | Amplitude         | 0.985       | 0.992      | 0.964      | 0.972       |
> > > | Pattern           | 0.971       | 0.996      | 0.977      | 0.970       |
> > >
> > > **Real-world Dataset Evaluation:** We further evaluate on the real-world Electricity dataset by examining LLM–human agreement via measuring the proportion of sample pairs where LLM preferences align with human annotations. To ensure fairness and credibility, the human labels were produced by an expert with practical experience in time-series analysis but not involved in the design or development of our method. The expert was asked to evaluate sample blocks based on the four criteria used in our framework, using intuitive qualitative standards. For example, trend was judged by the clarity of long-term directional changes, frequency by periodic repetition consistency, amplitude by the meaningfulness of fluctuation magnitude, and pattern by the recognizability of characteristic temporal structure. The results indicate considerable alignment, with LLM–human agreement rates mostly above 87%, suggesting that TSRating’s scoring is largely consistent with human understanding of time-series quality in realistic scenarios.
> > >
> > > | Quality Criterion | GPT-4o-mini vs Human | Gemini-2.0 vs Human | Claude-3.5 vs Human |
> > > | ----------------- | -------------------- | ------------------- | ------------------- |
> > > | Trend             | 0.9733               | 0.9254              | 0.8844              |
> > > | Frequency         | 0.9367               | 0.8984              | 0.8911              |
> > > | Amplitude         | 1.0000               | 0.9000              | 0.9242              |
> > > | Pattern           | 0.9041               | 0.8811              | 0.8736              |
> > >
> > > **Inter-LLM Agreement Evaluation:** We also assess inter-LLM agreement on the Electricity dataset, finding consistent preferences across different LLM pairs, suggesting that different LLMs tend to agree reasonably well:
> > >
> > > | Quality Criterion | GPT-4o-mini vs Gemini-2.0 | GPT-4o-mini vs Claude-3.5 | Gemini-2.0 vs Claude-3.5 |
> > > | ----------------- | ------------------------- | ------------------------- | ------------------------ |
> > > | Trend             | 0.9259                    | 0.9133                    | 0.9065                   |
> > > | Frequency         | 0.8917                    | 0.9657                    | 0.9000                   |
> > > | Amplitude         | 1.0000                    | 1.0000                    | 0.9091                   |
> > > | Pattern           | 0.9531                    | 0.8722                    | 0.9038                   |

---

> > > > ### Author Response · Authors · 2025-11-23
> > > >
> > > > > ##### [Continue] Re W2 & Q1 ....
> > > >
> > > > **Prompt Sensitivity Evaluation:** We conduct two new evaluations: 1) we evaluate the effect of swapping the order of examples and instructions within the prompt, with results suggesting little impact on judgments:
> > > >
> > > > | Quality Criterion | Original vs Order Swap |
> > > > | ----------------- | ---------------------- |
> > > > | Trend             | 1.0000                 |
> > > > | Frequency         | 1.0000                 |
> > > > | Amplitude         | 1.0000                 |
> > > > | Pattern           | 1.0000                 |
> > > >
> > > > 2) We evaluate the paraphrased prompt of the original prompt, which shows high agreement between the original and paraphrased prompts:
> > > >
> > > > | Quality Criterion | Original vs Paraphrase |
> > > > | ----------------- | ---------------------- |
> > > > | Trend             | 0.9888                 |
> > > > | Frequency         | 1.0000                 |
> > > > | Amplitude         | 1.0000                 |
> > > > | Pattern           | 1.0000                 |
> > > >
> > > > **Bradley–Terry Rating Robustness Evaluations:** We conduct a new experiment to evaluate the robustness of the Bradley–Terry rating under noisy pairwise labels by simulating increasing noise levels and measuring Spearman correlations of the resulting rankings. The ratings remain relatively stable under noise levels up to 8%, which is within a controllable range relative to the inherent inaccuracy of LLM judgments, suggesting moderate noise tolerance:
> > > >
> > > > | Noise Level | 0%    | 2%     | 4%     | 6%     | 8%     |
> > > > | ----------- | ----- | ------ | ------ | ------ | ------ |
> > > > | Correlation | 1.000 | 0.9854 | 0.9715 | 0.9496 | 0.9257 |
> > > >
> > > > **LLM Judgement Uncertainty Evaluation:** We evaluate the impact of uncertainty of LLM's judgement on time series patterns. While direct control over random seeds is limited in the API, we vary the LLM temperature parameter (0.1, 0.5, 1.0) on GPT-4o-mini’s Trend criterion judgments. The comparisons suggest strong consistency across these temperature settings:
> > > >
> > > > | Temperature Comparison | Agreement |
> > > > | ---------------------- | --------- |
> > > > | 0.1 vs 0.5             | 1.0       |
> > > > | 0.1 vs 1.0             | 1.0       |
> > > > | 0.5 vs 1.0             | 1.0       |
> > > >
> > > > In sum, these new experiments suggest that although LLM judgments are not perfectly stable or entirely free from bias, they provide adequate consistency and robustness across varying prompts, noise levels, and model variants within our experimental setup. That being said, we acknowledge that exploring uncertainty calibration, optimizing prompt design, and incorporating expert adjudication are important directions for future research. These considerations, however, do not diminish the significance of the present work, which demonstrates that LLMs offer a promising new direction for time-series rating.

---

> ### Author Response · Authors · 2025-11-23
>
> > ##### Re W3 & Q2: No ablation studies replacing or fine-tuning MOMENT encoder, raising concerns that gains may come from pretraining/domain overlap; does TSRater maintain performance with fine-tuned or alternative TSFM backbones?
>
> **Ablation Study on Different TSFMs:** We thank the reviewer for raising this important point. Following your suggestion, we conduct new ablation experiments on two different popular time series foundation models (TSFM), including Chronos and TimeGPT. Specifically, we perform data selection on the Weather dataset and evaluate the performance of five downstream models, using experimental settings consistent with those in the main experiments.
>
> |         | Linear | CNN   | PatchTST | iTransformer | TimeMixer |
> | ------- | ------ | ----- | -------- | ------------ | --------- |
> | MOMENT  | 0.611  | 0.734 | 0.467    | 0.444        | 0.433     |
> | Chronos | 0.613  | 0.735 | 0.464    | 0.447        | 0.428     |
> | TimeGPT | 0.609  | 0.738 | 0.479    | 0.438        | 0.435     |
>
> While the choice of TSFM does have some impact on downstream tasks, the effect is not significant based on our discovery, and we find that MOMENT can indeed be replaced with other TSFMs without substantial loss in performance.
>
> **Domain-Shift Leakage Evaluation:** We sincerely appreciate the reviewer’s insightful comment regarding domain-shift leakage. We conduct a new experiment to address this concern. Specifically, we note that the pretraining corpus of MOMENT (Time Series Pile) and the primary training corpus for Meta TSRater (Time 300B) come from distinct data domains. For instance, datasets like China_air (climate domain) and Wiki (web domain) used in TSRater training are not part of the Time Series Pile. Similarly, other TSFM encoders are trained on data with different domain distributions, which further supports the view that there is no significant domain overlap. As a result, we believe the observed performance improvements can be mainly attributed to our proposed rating scheme.
>
> Additionally, we conduct experiments on the Wiki dataset, which is not part of the MOMENT pretraining data, and observe that the performance of the downstream models remains strong, as shown below:
>
> |         | Linear | CNN   | PatchTST |
> | ------- | ------ | ----- | -------- |
> | Random  | 0.713  | 0.644 | 0.376    |
> | DataOob | 0.702  | 0.627 | 0.374    |
> | DataShapley | 0.700  | 0.633 | 0.374 |
> | KNNShapley  | 0.703  | 0.634 | 0.380  |
> | TimeInf     | 0.710  | **0.619** | 0.373  |
> | TSRating    | **0.697**  | **0.619** | **0.368**  |
>
> These results suggest that our approach maintains competitive performance on datasets outside the domain of the encoder pretraining, further reinforcing the effectiveness of our proposed rating scheme.
>
>
>
> > ##### Re W4: Selecting the top-50 percent across all methods may advantage rank-based raters.
>
> We thank the reviewer for raising this important point regarding the potential advantage of rank-based raters when selecting the top 50% of samples. We fully acknowledge the concern and have conducted additional experiments to address it.
>
> As part of our investigation, we conduct a data pruning experiment in which we progressively remove different proportions (ranging from 5% to 45%) of the highest-quality samples and observe the corresponding impact on the downstream model's performance. The results, shown in Figure 2, indicate that at various pruning ratios, TSRating shows a faster and more pronounced performance decline compared with other methods. This highlights TSRating’s ability to effectively identify and prioritize influential high-quality data. More details can be found in Section 4.2.
>
> Additionally, we have also evaluated a broader range of selection ratios. We evaluate TSRating's performance on the Traffic dataset using iTransformer under different selection ratios, and the RMSE results are summarized below. These results demonstrate that TSRating consistently outperforms competing methods at a majority of selection ratios, which further suggests that its advantages are not confined to selecting the top 50% samples but remain robust across various fractions of retained samples. More details can be found in Appendix D.6.
>
> | Method/Selection Ratio | 0.2   | 0.4   | 0.6   | 0.8   | 1.0   |
> | ---------------------- | ----- | ----- | ----- | ----- | ----- |
> | Random                 | 0.382 | 0.366 | 0.356 | 0.352 | 0.339 |
> | DataOob                | **0.371** | 0.364 | 0.354 | 0.346 | 0.339 |
> | DataShapley            | 0.381 | **0.356** | 0.353 | 0.343 | 0.339 |
> | KNNShapley             | 0.378 | 0.363 | 0.355 | 0.346 | 0.339 |
> | TimeInf                | 0.377 | 0.360 | 0.357 | 0.348 | 0.339 |
> | TSRating               | 0.372 | **0.356** | **0.349** | **0.342** | 0.339 |
>
> These findings suggest that TSRating's performance benefits are not solely dependent on the top 50% selection but are consistently observed across various sample selection thresholds.

---

> > ### Author Response · Authors · 2025-11-23
> >
> > > ##### Re W5: Runtime Table 2 lacks details on token usage, pairwise comparison budgets per dataset, and the impact of prompting policies on judgment quality.
> >
> > We thank the reviewer for raising this important and practical concern. In response, we conduct additional analyses to provide a clearer breakdown of the computational cost of “LLM judgments”. TSRater is trained using LLM pairwise comparisons collected from 22 datasets. Across all datasets, the total token usage is approximately **14.3 million tokens**, averaging around 0.65 million tokens per dataset. To avoid bias toward particular datasets or domains, a similar number of comparisons is allocated to each dataset. This corresponds to an average runtime of roughly **190 seconds** and a cost of about **$0.19 USD per dataset**. For reference, the overall runtime statistics from Table 2 are listed below:
> >
> > | Method / Component  | Time (seconds) |
> > | ------------------- | -------------- |
> > | DataShapley*        | 210,000        |
> > | KNNShapley          | 152            |
> > | DataOob             | 4,785          |
> > | TimeInf             | 4,938          |
> > | TSRater (Total)     | 4,687          |
> > | – LLM Judgments     | 4,167          |
> > | – Meta-training     | 323            |
> > | – Few-shot Tuning   | 55             |
> > | – Inference         | 142            |
> > (* indicates estimates from prior work)
> >
> > We also examine the effects of prompting design on judgment quality. All prompts used in our experiments are listed in Appendix A and follow common LLM evaluation practices, including explicit task descriptions, structured answer formats, and example demonstrations for each quality dimension. To assess robustness, paraphrased versions of the prompts are tested (as discussed in W2), and the results show high agreement compared with the original versions. This suggests that the judgments are not overly sensitive to surface-level variations in wording.
> >
> > To further investigate the importance of in-prompt examples, we conduct an ablation study removing these examples. On the synthetic dataset, judgment accuracy of GPT-4o-mini shows a measurable decrease when examples were omitted:
> >
> > | Criterion   | With examples | Without examples |
> > | ----------- | -------------- | ---------------- |
> > | Trend       | 0.9525         | 0.9250           |
> > | Frequency   | 0.9625         | 0.9575           |
> > | Amplitude   | 0.9850         | 0.9625           |
> > | Pattern     | 0.9725         | 0.9450           |
> >
> >
> > > ##### Re W6: Labeling synthetic blocks with obvious hand-crafted time series features as high-quality is is not make sense (see W1).
> >
> > We thank the reviewer for this helpful observation. Real-world time series indeed contain noise, irregular structures, and unexpected behaviors, and describing the synthetic samples with “obvious hand-crafted time series characteristics” as “high-quality blocks” may unintentionally give the impression that such patterns are common or typical in practice. We will revise the description in Line 246 to avoid this potential misunderstanding.
> >
> > As a clarification, the synthetic dataset (as discussed in line 246) was originally designed as a simplified and controlled experiment, where the presence or absence of specific characteristics (e.g., trend, seasonality, amplitude stability, or recurring patterns) can be clearly defined. Our goal was to provide a clean setting to examine whether the LLM can detect the types of signals that TSRating considers. In addition, this setting also facilitates human inspection by providing clearer visual interpretations when analyzing and examining the experimental results. However, our experiment setting does not intend to suggest that real-world series naturally exhibit such idealized structures.
> >
> > In practical usage, TSRating operates through relative pairwise comparison within each dataset. Instead of assuming that certain shapes represent globally “good” or “bad” time series, it orders samples according to conformity to the four defined criteria. The ordering is relative rather than absolute, and intended to adapt to the characteristics of each dataset. To further support this, we also provide visualizations of high- and low-ranking samples on real datasets, with details shown in Figure 4-6.
> >
> >
> > > **Additional References**
> >
> > [1] Schreiber T. Determination of the noise level of chaotic time series[J]. Physical Review E, 1993, 48(1): R13.
> >
> > [2] Cai Y, Choudhry A, Goswami M, et al. Timeseriesexam: A time series understanding exam[J]. arXiv preprint arXiv:2410.14752, 2024.

---

> > > ### Comment · Reviewer_nhuS · 2025-11-27
> > >
> > > I appreciate the authors' efforts and their detailed response, which have addressed my concerns. Accordingly, I have revised my rating to recommend acceptance.

---

> > > > ### Author Response · Authors · 2025-11-27
> > > >
> > > > We sincerely thank you for your detailed review and valuable suggestions. Your insightful feedback has been instrumental in improving our work. We truly appreciate the time and effort you invested in evaluating our paper, and we’re very pleased that our response has addressed your concerns.

---

### Author Response · Authors · 2025-12-03
**Summary of Rebuttal**

Dear Area Chairs, Senior Area Chairs, and Program Chairs,

Thank you for overseeing the review process of our submission. As the rebuttal period comes to a close, we would like to sincerely thank the reviewers for their valuable feedback and provide a concise summary of the main points discussed. Below, we first summarize the positive feedback and key strengths highlighted by the reviewers, then provide overall responses to the major concerns raised, and finally present the rebuttal status and score changes to facilitate review tracking.

## Paper Overview and Strengths

Our paper proposes TSRating, an innovative framework for evaluating the quality of diverse time series datasets by leveraging large language models (LLMs) as general “pattern evaluators” combined with meta-learning techniques. The method effectively addresses the critical challenge of time series data quality assessment, which is essential for robust deep time series model training. Reviewers have positively recognized several key strengths of our work:

- The paper is well-organized, clearly written, and easy to follow, with logical presentation of the problem, methodology, and results. (Reviewer nhuS, Reviewer pUfp)
- The proposed TSRating framework innovatively leverages large language models (LLMs) as general “pattern evaluators,” moving beyond traditional statistical or contribution-based metrics to capture complex time series characteristics. (Reviewer kv5d)
- The meta-learning approach, trained across 9 domains and 22 diverse datasets, effectively addresses domain heterogeneity and enables strong cross-domain generalization, making it practical for unseen datasets. (Reviewer nhuS, Reviewer kv5d)
- The combination of techniques, including pairwise comparisons, Bradley-Terry modeling, TSFM encoder, and MAML with SignSGD, is thoughtfully designed and experimentally validated across multiple datasets and tasks, demonstrating both novelty and solid empirical support. (Reviewer pUfp)
- Our framework offers a scalable, adaptable solution to automated data quality assessment, filling an important gap in time series analysis and providing measurable improvements in downstream applications. (Reviewer pUfp, Reviewer kv5d)

---

## Summary of Reviewer Concerns and Our Responses

### 1. Applicability and Limitations of TSRating to Noisy and Irregular Time Series

Reviewer nhuS and Reviewer kv5d noted that TSRating assumes series with strong trends, seasonality, and regular patterns are of high quality, which may limit applicability to real-world scenarios featuring noise, irregularities, or important anomalous patterns. They also pointed out that synthetic datasets with hand-crafted ideal patterns might not reflect real-world complexity.

**Response:** We conducted extensive new experiments on a challenging real-world Weather dataset with higher noise and irregular fluctuations. TSRating improves downstream forecasting performance over alternatives, confirming applicability beyond idealized data. We clarified that TSRating’s scoring provides a meaningful ordering from regular to irregular samples, enabling use cases both for selecting high-quality regular samples and identifying irregular/noisy or anomalous samples by focusing on low-rated or extended criteria. We demonstrated this flexibility with a new case study on anomaly detection, adding an explicit anomaly criterion, where TSRating outperforms baseline methods. Figures 1 and 4 were revised or expanded to include more realistic, noisy samples to avoid misleading impressions. These updates are detailed in **Appendix B** and **Appendix F**.

### 2. Robustness, Consistency, and Bias of LLM Judgments

Reviewer nhuS, Reviewer kv5d and Reviewer pUfp raised concerns about LLM judgment biases (e.g., pattern or positional bias), lack of thorough validation, inter-LLM agreement, and uncertainty analysis. Reviewer nhuS suggested examining stability across prompts, seeds, and different LLMs.

**Response:** We acknowledge that LLM-based judgments inevitably contain certain forms of bias. To examine the reliability of TSRating under such conditions, we expanded controlled evaluations on both synthetic and real-world datasets using multiple LLMs. Results show high alignment with human judgments (above 87%) and strong agreement across different LLMs (above 92%). We further tested stability under prompt paraphrasing and input order swapping, observing minor effects on judgment results. Additionally, we assessed the robustness of the Bradley–Terry model under noisy labels and evaluated uncertainty through temperature variation experiments, confirming moderate noise tolerance and consistent performance. We have added further evaluations in **Appendix G**.

---

> ### Author Response · Authors · 2025-12-03
> **Summary of Rebuttal [continue]**
>
> ### 3. Justification and Adaptability of the Four Quality Criteria
>
> Reviewer kv5d and Reviewer pUfp questioned whether the four criteria (trend, frequency, amplitude, pattern) sufficiently and objectively capture diverse time series characteristics, especially for tasks requiring irregular data.
>
> **Response:** The four criteria are grounded in classical theory and work well for many forecasting and classification tasks. Furthermore, we conducted ablation experiments removing each criterion individually, and the results consistently show performance drops across multiple datasets, demonstrating that all four criteria contribute meaningfully to TSRating under the current formulation. To handle irregular and anomalous data, we extended TSRating by introducing a new anomaly detection criterion, demonstrating improved performance in anomaly detection tasks. This extension illustrates TSRating’s flexible design that can incorporate additional, task-specific criteria. We also acknowledged that exploring more dynamic and task-adaptive criteria is a valuable direction for future work. These discussions and extensions are detailed in **Section 3.3** and **Appendix F**.
>
> ### 4. Sensitivity of Time Series Foundation Model (TSFM) Encoders
>
> Reviewer nhuS and Reviewer pUfp pointed out a lack of ablation on different TSFM encoders and raised concerns about domain overlap or leakage between encoder pretraining and meta-training data.
>
> **Response:** We performed ablation experiments with alternative TSFMs (Chronos, TimeGPT) showing comparable downstream results, indicating TSRating may not be dependent on a specific encoder. We confirmed that encoder pretraining corpora and TSRating training datasets are disjoint in domain, reducing concerns of domain leakage. Additional tests on out-of-domain datasets like Wiki further validate the approach’s robustness. The ablation experiments are detailed in **Appendix D.6**.
>
> ### 5. Adaptive Aggregation for Criteria and Channels
>
> Reviewer kv5d and Reviewer pUfp suggested exploring more adaptive aggregation methods beyond simple averaging for both the four criteria and multi-channel inputs.
>
> **Response:** We agree that adaptive aggregation is a valuable direction. To examine its potential, we added new experiments incorporating weighted aggregation across both criteria and channels based on domain knowledge (Weather dataset). The results show that the performance differences are modest, indicating that simple averaging is already an effective choice in the current setting. Beyond this experiment, we also plan to explore more adaptive and learnable aggregation strategies. Instead of relying on manually defined weights, future work could allow both criterion- and channel-wise weights to be optimized during meta-rater training, potentially using gating mechanisms similar to Mixture-of-Experts. The discussions and experiments about criteria aggregation have been added in **Appendix E**.
>
> ### 6. Other Improvements and Revisions
>
> We also made several additional enhancements to improve completeness and clarity, including:
>
> - Providing detailed token usage of LLM judgements collection in Table 2
> - Adding more real selected sample visualizations in Appendix B.2
> - Supplementing Figure 3 with additional fine-tuning experiments using another LLM MOMENT
>
> We have uploaded the revised PDF with all modifications and additions marked in **blue font** for easy review.
>
> ---
>
> ## Rebuttal Status and Score Updates
>
> We are sorry to learn about the recent data-leakage incident affecting the OpenReview platform. To help reduce the workload for the newly assigned Area Chairs and facilitate a smooth reassessment, we provide a concise summary of the rebuttal-stage status for our submission.
>
> | Reviewer ID | Initial Score | Score After Rebuttal | Confidence (unchanged) | Last Response Timestamp (EST) | OpenReview Official Announcement Time (EST) |
> | ----------- | ------------- | -------------------- | ---------------------- | ----------------------------- | ------------------------------------------- |
> | nhuS        | 4             | 6                    | 5                      | 2025-11-27, 00:41AM           | 2025-11-27, 10:09AM                         |
> | kv5d        | 4             | 6                    | 3                      | 2025-11-26, 02:47AM           | 2025-11-27, 10:09AM                         |
> | pUfp        | 6             | 8                    | 4                      | 2025-11-27, 00:36AM           | 2025-11-27, 10:09AM                         |
>
> We would like to clarify that we have never accessed or utilized any leaked data during the review process.
> All reviewer acknowledgements, including both for addressed concerns and for evaluation score increases, occurred at least **9 hours before** the OpenReview identity leakage announcement time (i.e., 10:09AM, 11-27 EST). We became aware of the OpenReview incident only after the final reviewer responses were submitted (i.e., 00:41AM, 11-27 EST).

---

> > ### Author Response · Authors · 2025-12-03
> > **Summary of Rebuttal [continue]**
> >
> > We sincerely thank all the reviewers for their thoughtful, constructive feedback and valuable suggestions that have greatly helped improve the quality and clarity of our work. We also deeply appreciate the Area Chair’s dedicated efforts in managing the review process and facilitating a fair, thorough evaluation of our submission. Thank you all for your time and commitment to maintaining high standards in the review.

---

### Meta-Review · Area_Chair_sdMx · 2025-12-22

**Summary:**

The paper presents TSRating, a unified framework for rating time-series data quality across diverse domains. TSRating first uses LLM prompting to collect pairwise quality judgments, then trains a lightweight predictor (TSRater) to efficiently score new samples. To improve cross-domain adaptability, it applies meta-learning over comparisons drawn from nine domains, using signSGD for efficient inner-loop updates. Experiments across 11 datasets and multiple time-series tasks (with both conventional models and foundation models) show improved accuracy, efficiency, and robustness over existing baselines.

Reviewers raised several important concerns in the initial reviews, including: (1) potential bias in LLM-based judgments, (2) generalization to noisy and irregular time series, (3) justification for the four proposed quality criteria, and (4) missing or insufficient ablations.

The rebuttal addresses these major issues comprehensively with clarifications and additional evidence. As a result, the paper’s overall presentation and technical case are strengthened, and reviewers have converged toward a consensus in support of acceptance.

**Reviewer Concerns:**

All major concerns summarized in the meta-review have been adequately addressed in the rebuttal.

**Reviewer Scores:**

Three reviewers indicated that the rebuttal addressed their main concerns and agreed to raise their scores prior to the OpenReview incident.

---

### Decision · Program_Chairs · 2026-01-26

Accept (Poster)